# A cancer drug atlas enables synergistic targeting of independent drug vulnerabilities

Ravi S. Narayan[1,17], Piet Molenaar[2,17,18], Jian Teng[3,17], Fleur M. G. Cornelissen[4], Irene Roelofs[4], Renee Menezes[5], Rogier Dik[1], Tonny Lagerweij[4], Yoran Broersma[4], Naomi Petersen[4,6], Jhon Alexander Marin Soto[4,7], Eelke Brands[1], Philip van Kuiken[4], Maria C. Lecca[8], Kristiaan J. Lenos[8], Sjors G. J. G. In 't Veld[4], Wessel van Wieringen[9], Frederick F. Lang[10], Erik Sulman[11], Roel Verhaak[12], Brigitta G. Baumert[13], Lucas J. A. Stalpers[14], Louis Vermeulen[8], Colin Watts[15], David Bailey[16], Ben J. Slotman[1], Rogier Versteeg[2], David Noske[4], Peter Sminia[1], Bakhos A. Tannous[3], Tom Wurdinger[4], Jan Koster[2,19] & Bart A. Westerman[4,19 ✉]

Personalized cancer treatments using combinations of drugs with a synergistic effect is attractive but proves to be highly challenging. Here we present an approach to uncover the efficacy of drug combinations based on the analysis of mono-drug effects. For this we used dose-response data from pharmacogenomic encyclopedias and represent these as a drug atlas. The drug atlas represents the relations between drug effects and allows to identify independent processes for which the tumor might be particularly vulnerable when attacked by two drugs. Our approach enables the prediction of combination-therapy which can be linked to tumor-driving mutations. By using this strategy, we can uncover potential effective drug combinations on a pan-cancer scale. Predicted synergies are provided and have been validated in glioblastoma, breast cancer, melanoma and leukemia mouse-models, resulting in therapeutic synergy in 75% of the tested models. This indicates that we can accurately predict effective drug combinations with translational value.

[1] Department of Radiation Oncology, Amsterdam UMC, location VUMC, Cancer Center, Amsterdam, the Netherlands. [2] Department of Oncogenomics, Academic Medical Center, Amsterdam, the Netherlands. [3] Experimental Therapeutics and Molecular Imaging Lab, Neuroscience Center, Neuro-Oncology Unit, Massachusetts General Hospital and Harvard Medical School, Boston, MA, USA. [4] Department of Neurosurgery, Amsterdam UMC, location VUMC, Cancer Center, Amsterdam, the Netherlands. [5] Department of Psychosocial Research and Epidemiology, Netherlands Cancer Institute, Amsterdam, the Netherlands. [6] Department of Medical Oncology, Amsterdam UMC, location VUMC, Cancer Center, Amsterdam, the Netherlands. [7] Department of Hematology, Amsterdam UMC, location VUMC, Cancer Center, Amsterdam, the Netherlands. [8] Center for Experimental Molecular Medicine (CEMM), Laboratory for Experimental Oncology and Radiobiology (LEXOR), Amsterdam UMC, location AMC, Cancer Center, Amsterdam, the Netherlands. [9] Department of Epidemiology and Biostatistics, Amsterdam UMC, location VUMC, Amsterdam, the Netherlands. [10] Department of Neurosurgery, Division of Surgery, The University of Texas MD Anderson Cancer Center, Houston, TX, USA. [11] Department of Radiation Oncology, NYU Langone Health, New York, NY 10016, USA. [12] Department of Computational Biology, The Jackson Laboratory for Genomic Medicine, Farmington, CT, USA. [13] Department of Radio-Oncology, Kantonsspital Graubünden, Loëstrasse 170, CH-7000 Chur, Switzerland. [14] Radiation Oncology, Academic Medical Center, Amsterdam, the Netherlands. [15] Neurosurgery, Institute of Cancer and Genomic Sciences, University of Birmingham, Edgbaston, Birmingham B15 2TT, UK. [16] IOTA Pharmaceuticals Ltd, St Johns Innovation Centre, Cowley Road, Cambridge, UK. [17] These authors contributed equally: Ravi S. Narayan, Piet Molenaar, Jian Teng. [18] Deceased: Piet Molenaar. [19] These authors jointly supervised this work: Jan Koster, Bart A. Westerman. ✉email: a.westerman@amsterdammumc.nl

Personalized therapies against tumor-driving targets are being used effectively in the clinic, but in many cases drug resistance occurs giving rise to inevitable relapses[1–7]. Since tumors are dependent on a limited number of molecular mechanisms for their survival/proliferation, combination therapy enables simultaneous targeting of these crucial mechanisms and is expected to decrease therapy resistance[8–12].

Many positive effects of drug combinations in the clinic are reflective of the best response to either one of the two drugs[13]. Therefore, combinations of drugs are commonly more effective because each drug compensates for the drawback of the other drug. Currently, only a fraction of these combinations provide synergistic (i.e., more than additive) effects[13]. Therefore, the identification of crucial mechanisms that lead to synergistic drug effects is highly desirable. However, the identification of these drug combinations has so far only been possible using an empirical setting (i.e., high-throughput testing of all combinations for each cell line), followed by identification of molecular features such as genetic mutations and transcriptome, methylome, and proteome genomic data to predict the therapy response (reviewed[14–16]). This showed that synergy prediction is possible[17], although with a limited overall probability and only applicable on a defined lineage background. The recent pan-cancer DREAM community effort of the Drug Combination Prediction Challenge confirmed these previous findings on a pan-cancer scale[18].

Drug combination therapies where drugs work synergistically are expected to be particularly useful for tumor types for which chemotherapeutic and targeted approaches have failed or show frequent cases of therapy resistance. Among these are Glioblastoma (GBM) but also triple-negative breast tumor patients[19], BRAF-driven melanoma[20], and BCR-ABL-driven chronic myeloid leukemias[21].

Given that not all possible drug combinations can be tested onto each patient-specific mutation profile, a major challenge is to orchestrate the most effective combination therapies to a large range of genetic mutation profiles of patients. Here, we use an approach to identify synergistic drug pairs based on a method that we call the drug atlas. This method enables us to predict drug vulnerabilities based on single drug-response data on a pan-cancer scale and link this to personalized features. Our methodology forms a generalizable strategy to identify personalized multidrug therapies and enabled us to identify novel and unexpected combination therapies.

## Results

**A drug atlas allows visualization of complex synergistic drug interactions**. The number of possible effective combinations of existing anticancer drugs is enormous and calls for a rational approach to select the most potential combinations, also taking into account the genetic background of the individual case. We reasoned that relations between cancer processes can be reflected by the relations between drug effects. Therefore, drug-response data might guide us towards combination therapies that affect tumor-driving processes simultaneously. Our approach is exemplified in Fig. 1a: if a cell line is sensitive to drug A and not to drug B, or vice versa, then the underlying processes are apparently working independently by showing exclusive vulnerabilities. If a third cell line is, however, sensitive to both drug A and B, then these independent processes can be targeted simultaneously and form a co-vulnerability. We argued that we can use this concept to identify commonly occurring co-vulnerabilities in cancer cell lines, and when these are treated with the right combination therapies, then more than additive (synergistic) drug effects can be expected.

To determine which processes work independently, we calculated the level of dissimilarity of single-drug effects over many cell lines. For this, we used 60,000 previously published drug dose–response curves obtained from the Sanger GDSC1000 and Novartis/Broad CCLE drug-encyclopedias (all sources, including hyperlinks, are summarized in Supplementary Data 1). The area under the curve from these data, representative for the drug effect, was subsequently clustered using Ward or average hierarchical clustering (details are provided in the Methods section). Drugs that have similar responses over many cell lines will end up in the same cluster while drugs that have dissimilar responses over many cell lines will cluster relatively further away. To visualize these cluster-distance relations, the cluster tree was projected as a 2D Voronoi diagram which we call the drug atlas. This drug atlas provides an intuitive overview of drug-effect relations over many cell lines. The atlas method was validated in parallel using world map coordinate relations (Supplementary Fig. 1a, see also Supplementary Fig. 1b–d).

To investigate how drug combinations with a synergistic effect relate to drug-effect similarities, we curated all published and peer reviewed synergy data matching the cell line data that were used to create the drug atlas. This resulted in identification of 483 drug pairs that showed a synergistic effect in 156 cell lines (Supplementary Data 2, references are given in the Supplementary References). These synergistic interactions are visualized onto the drug atlas by drawing a line between the respective drugs (Fig. 1b, a more detailed view on drug targets[22] is given in Supplementary Fig. 1e). An example from the list of curated data is shown in Fig. 1c.

Consistent with our concept, most synergistic drug interactions span a large distance on the atlas, showing in qualitative way that the corresponding drug pairs affect unrelated processes.

**Drug distance, drug sensitivity, and targeted therapy correlate to drug synergy**. Based on our concept, we expect that inhibition of unrelated processes will result in synergistic effects since they represent independent survival mechanisms. To quantify this, we calculated the level of dissimilarity of drug effects using the drug distance. For this, we selected the GDSC (MGH) data[22], which showed the most consistent clustering (unlike the combined MGH and GDSC1000 data, Supplementary Fig. 1f, see also Haibe-Kains et al.[23]). As independent benchmark data, we used the DREAM drug-synergy challenge data, consisting of 11,173 synergy measurements[24]. Drug effects were already clustered to generate the drug atlas and were used to calculate the cophenetic distances between the clusters. The larger the cophenetic distance, the more unrelated the drug effects and the higher the drug distance is. As a reference, the drug distance of the full spectrum of possible interactions between all drugs was used as well as the distance within known pathways/gene ontologies[25]. In agreement with our concept, the average drug distance of drug pairs with synergistic effect significantly exceeded the average overall drug distance Fig. 2a, $P = 4 \times 10^{-4}$), confirming our initial hypothesis. Other distance models (average clustering) showed a similar outcome (see Supplementary Fig. 2a). Similarly as we determined the drug distance, we could calculate the target distance since every drug has a defined target[22]. For this, we used the already clustered drugs and then calculated the cluster distance based on the average of each target (since more drugs have the same target). This also showed that target distances of drug combinations with a synergistic effect commonly exceeded the average target distance of all possible pairs for the GDSC data (Fig. 2b, $P = 4.5 \times 10^{-15}$), which was confirmed for another clustering method (Supplementary Fig. 2b) and for the DREAM data (Fig. 2c) as well. The identified distance–synergy relationship

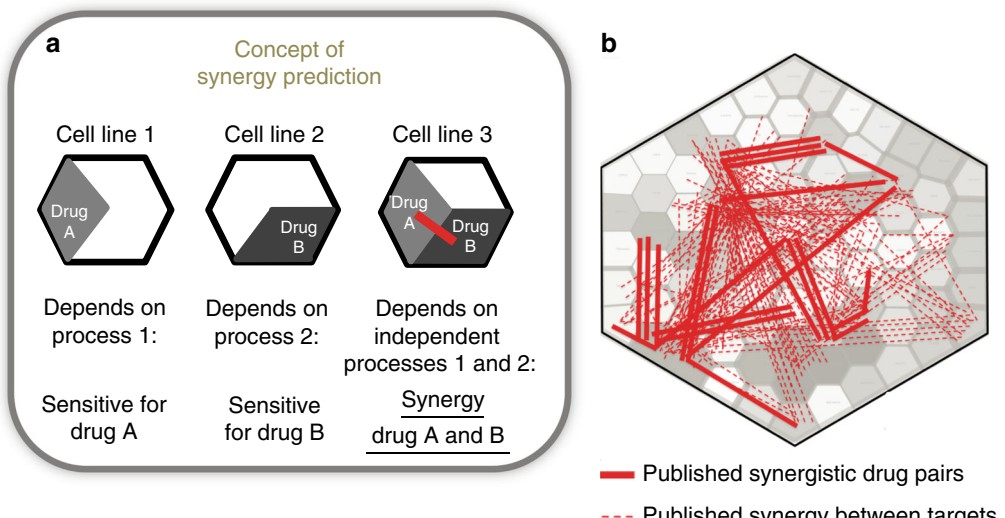

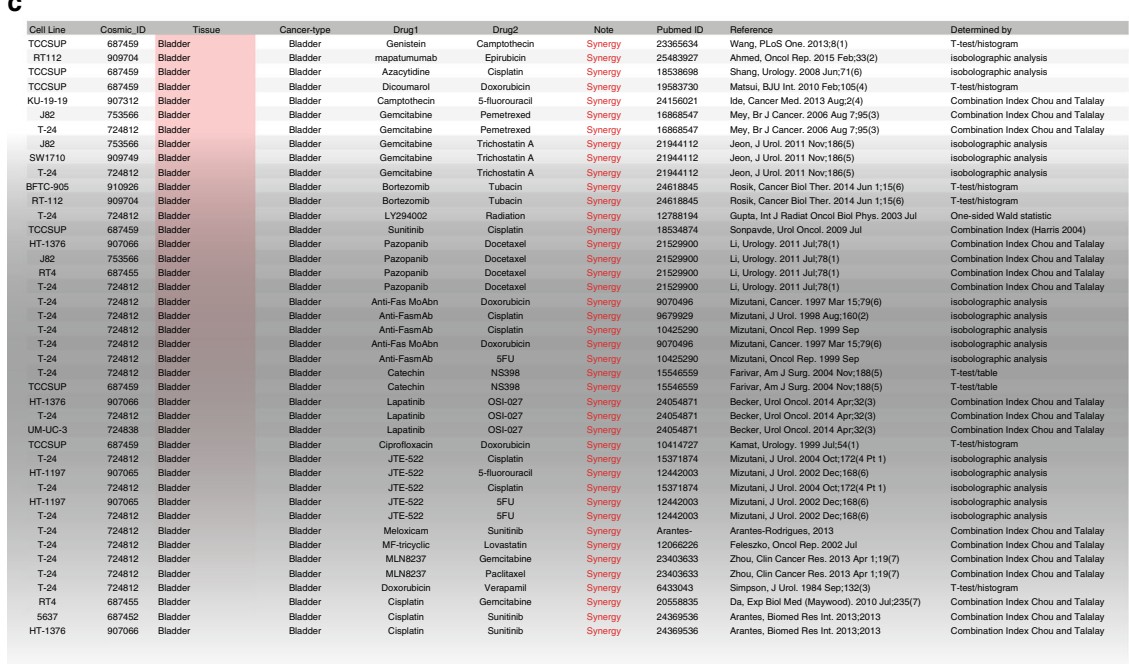

**Fig. 1 Concept of synergy prediction and depiction of the curated data on the atlas. a** Schematic representation of drug-atlas approach. If two cell lines have mutually exclusive sensitivity to either drug A or B, as shown by the intensity of each diamond, then two independent molecular mechanisms might be causal of this. However, when a third cell line is sensitive to both of these drugs, then these unrelated mechanisms are affected simultaneously, giving rise to a synergistic effect. Hence, in our model, synergy between drugs is expected when pairs of drugs act on independent processes. **b** Relations of drug dose–response data are difficult to comprehend, given the enormous amount of data points associated with them. We used a Vonoroi diagram to depict drug dose–response data of the GDSC drug–response encyclopaedia[29], which resulted in a drug atlas. Depicting manually curated drug synergies for the GDSC cell lines (483 synergistic drug pairs occurring in 156 cell lines), shows that synergistic drug pairs commonly have a distal location, as shown by synergistic drug–drug combinations (solid red lines) or synergistic target–target combinations (dotted lines). **c** An exempt of the list of curated synergy pairs that are matched to the GDSC data. The full list is available in Supplementary Data 2, and references are listed in the Supplementary References. In total, there are $n = 274$ cases of drug and target synergy shown on the drug atlas.

was independent of the within-pathway versus between-pathway distance[25], supporting the concept that distance, whether within or between known pathways, determines the chance of observing synergy. Consistently, given that most (80%) possible interactions occur between processes, synergies are mostly found between processes (Fig. 2d; Supplementary Fig. 2c).

Since we identified synergistic drug pairs for cell lines that are present in the GDSC data, we were able to match drug-sensitivity data to the cell lines that showed synergy in that

particular case. From our model, we expect a higher sensitivity in case synergy occurs. We therefore tested whether the occurrence of synergy correlates to drug sensitivity. This analysis indeed showed that the cell lines that show synergy with particular drugs were significantly more sensitive to these respective drugs than control cell lines from the matching tumor type (Fig. 2e, $P_{\text{overall}} = 1 \times 10^{-4}$, Supplementary Fig. 2d, $P = 6 \times 10^{-4}$). Similar results were obtained by using the DREAM dataset ($P < 1 \times 10^{-4}$, Fig. 2f).

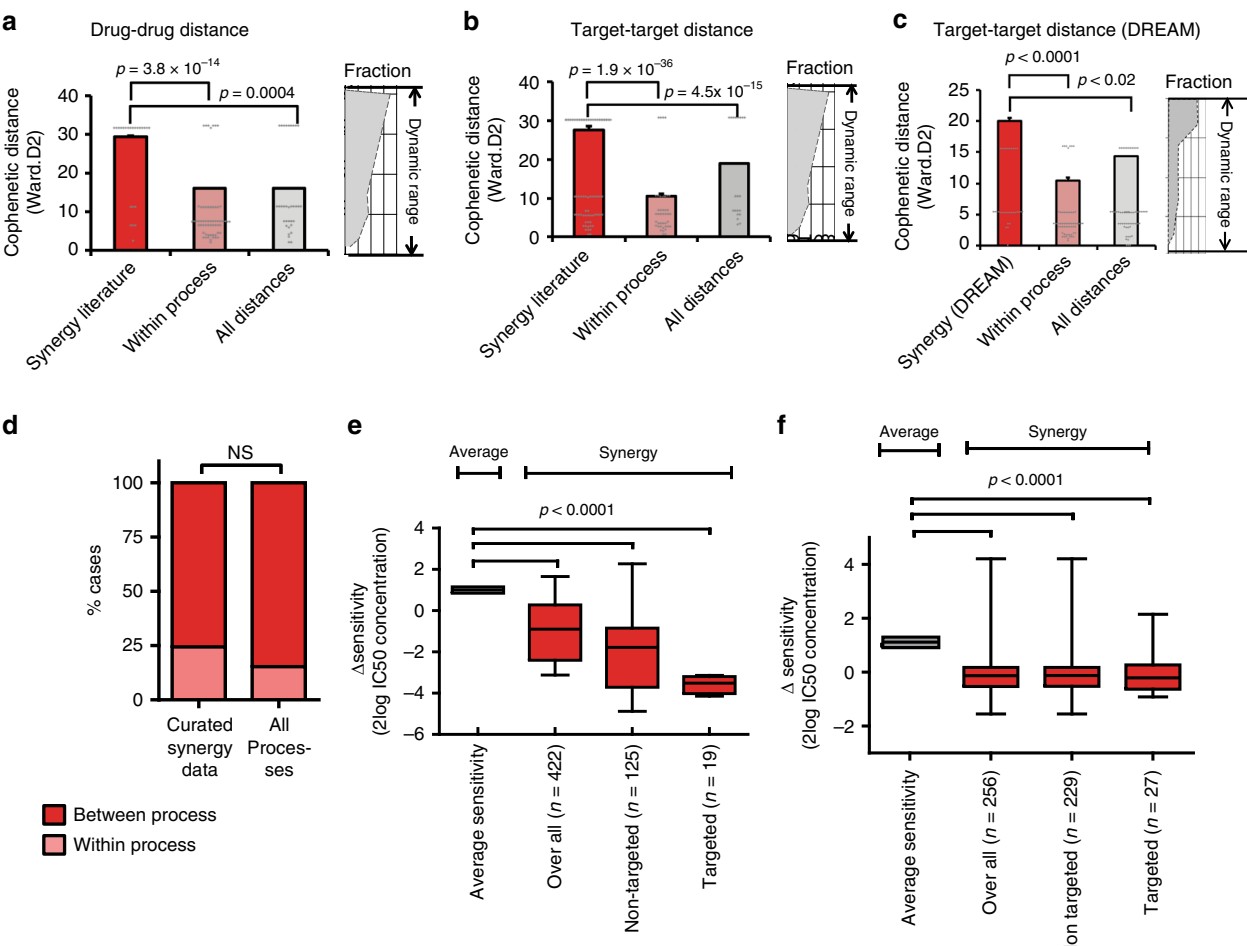

**Fig. 2 Synergistic drug pairs show a large distance on the drug atlas. a** The cophenetic distance (to quantify the drug-effect-dissimilarity) between the curated synergistic drug pairs was compared with distances between drugs in the same-ontology group or the distance between all drugs. The distance of curated synergistic drug pairs significantly exceeds the average distance between all drugs as well as the same-ontology[24] distance, which indicates that most synergistic drug pairs have a relative large drug distance. To calculate the cophenetic distance, WARD.D2 clustering was used (dynamic window shown on the right of the histogram). **b** Similar results were obtained when the distances between targets[29] of the drugs were used. **c** When the benchmark data of DREAM[24] were analyzed, similar results were seen when the distances between targets of the drugs were used. **d** Histograms showing that between-process interactions as seen in synergistic combinations match between-process interactions over all drug pairs, indicating that synergy occurs both within as well as between processes and is not limited to between-process interactions. **e** According to our model, sensitivity for both drugs is necessary for synergy to occur. Since we have used GDSC cell lines for our curation, we were able to match drug sensitivities to occurrence of synergy which showed that a significant higher sensitivity is observed for synergistic drugs compared with the overall sensitivity for the corresponding drugs. Sensitivities are normalized to 1 representing the average of all IC50s for a drug in a certain tissue. Overall sensitivity includes all known $IC_{50}$ values for the cases where synergy was observed. Targeted indicates that a targeted drug is used in a cell line that harbors the respective mutated target. Non-targeted indicates that mutation-targeted drugs in a non-mutated or non-amplified context. **f** When the benchmark data of DREAM[24] were analyzed, similar results were seen when the sensitivities of the drugs were used. *P*-values **a–f**, Student's *t* test (one-sided). Error bars histograms, standard error; box-and-whiskers plot, minimum, 25th percentile, median, 75th percentile, and maximum. Curated drug–drug distances synergistic drug pairs $n = 81$, all drug–drug distances: $n = 8515$; within-pathway distances, $n = 235$; target–target synergistic pairs $n = 193$. Comparison between and within pathways for all versus synergistic drug pairs: all within $n = 495$; all between $n = 2746$; synergy within $n = 117$; synergy between $n = 363$.

Since the cell lines used to build the drug atlas have been characterized on a genetic level by Novartis/Broad CCLE and Sanger GDSC1000 which is matched to drug-sensitivity profiles (confirmed by others[26,27]; CCLE data is now fully available[28]), we could analyze the role of mutations in relation to drug sensitivity upon presence of synergy, which showed that high sensitivity significantly corresponded to the presence of driver mutations, both by direct targeting of the protein, linking our synergy model to driver mutations (Fig. 2e, f, indicated by 'targeted'). Also, when cases of matched tumor drivers and their targeting drugs were excluded, a significantly increased sensitivity was seen for synergistic drugs (Fig. 2e, f, indicated by 'non-targeted'). For breast tumors, synergy with HER2-Neu/EGFR inhibitors

correlated significantly with HER2-Neu/EGFR-activating mutations ($P = 1 \times 10^{-4}$, Fig. 3a).

**Drug distance and sensitivity can predict synergy linked to tumor-driving mutations.** To independently provide evidence for our hypothesis, we performed an independent drug-synergy screen in GBM cell lines. For this, drugs were chosen to have a high drug distance as well as a high sensitivity in GBM cell lines (Supplementary Figs. 1d and 3a (showing sensitivity and distance, respectively)). We tested a total of 30 different combinations (Supplementary Data 1) onto 9 glioblastoma cell lines (which are part of the GDSC dataset). Drugs were titrated up to an IC50

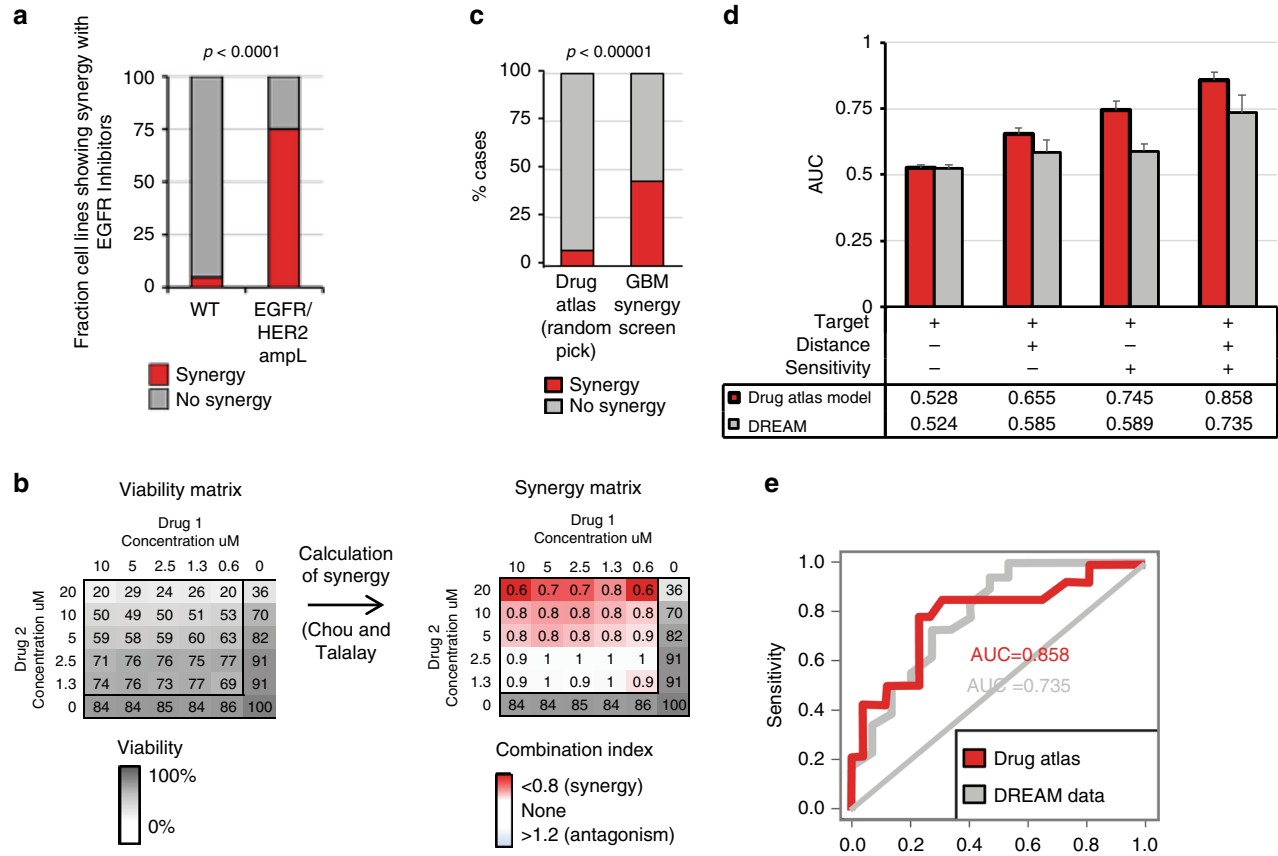

**Fig. 3 Synergistic drug pairs show a large distance on the drug atlas. a** Occurrence of synergy with EGFR/HER2 inhibitors in breast tumor cell lines is significantly linked to mutations of the EGFR or HER2 genes. **b** Example of a heatmap of a typical experimental result of our drug screen showing the relative viability as a result of the titration of two drugs in different combinations. Synergy was calculated by the median effect principle by Chou and Tallalay[29]. **c** A total of 30 preselected drug pairs were validated for synergistic efficacy in nine GBM cell lines. Drugs were chosen because these drugs showed a high drug distance on the drug atlas (see Supplementary Fig. 3a) and because they individually show a high sensitivity (see Supplementary Fig. 3b). The histogram shows the summary of the results of in vitro measurement of drug–drug synergy showing a significant enrichment over the background. **d** Area under the curve analysis of the dose-responder curve shows that the distance as well as the sensitivity contribute to the predictive power of the synergy prediction model for both tested datasets. **d** The synergy prediction model that we developed based on the previous data shows a good performance by the receiver operator curve analysis. Model performance was tested through cross-validation of the curated data and on the benchmark data of Menden et al.[24] and quantified using the area under the curve. P-values **a**, **c** $\chi^2$ test. Error bars histograms, standard error; Fraction of cell lines with EGFR/HER mutations ($n = 35$) is compared with wild-type cell lines ($n = 43$). The glioblastoma synergy screen was performed in triplicate and showed $n = 91$ (synergy) versus $n = 116$ (no synergy) as compared with random pick $n = 16$ (synergy) versus $n = 184$ (non-synergy). All dose–response effects were cross-validated numerous times. The prediction model was trained on 463 combinations, where the controls were taken iteratively ($n = 1000$). For the in vitro validation of synergy, non-consistent results were repeated until consistent.

(Supplementary Fig. 3b, Supplementary Data 3a), and the synergy between drugs was determined by studying the combined drug effect in a $6 \times 6$ matrix where each drug was titrated using a twofold dilution in each step (Fig. 3b). The viability was measured using CellTiter Glo 3D after 72 h of exposure to the drugs as examined in triplicate experiments. Based on these viabilities, the combination index was calculated using the median effect principle by Chou and Tallalay[29]. A substantial number of tested combinations showed synergy, i.e., up to 18 out of 30 pairs (60%) showed synergy over multiple cell lines and 116 out of 270 drug pairs (43%) showed synergy over all cell lines (i.e., having a combination index less than 0.8). Based on the curated synergy data, we could set a threshold that distinguishes synergistic drug pairs from randomly chosen drug pairs. Around 8% of randomly picked drug combinations meet these synergy criteria, indicating that we have a strong enrichment over the background (Fig. 3c, $P < 1 \times 10^{-5}$). The full list of determined combination therapies is given in Supplementary Data 3b. Chou and Talalay Combination Index synergy significantly correlated ($P < 1 \times 10^{-4}$) to other

synergy/additivity metrics (Loewe, BLISS or HSA method), see Supplementary Fig. 3c and Supplementary Data 3c, where each model interprets weak interactions differently (Supplementary Fig. 3d). Improvement of the interpretation of synergy data could become more robust by taking more complex interactions into account according to Wicha et al.[30]).

We generated a synergy predictor based on the (1) the individual drug sensitivities, (2) their target information, and (3) the drug distance (see Methods and Code Availability). We analyzed whether the sensitivity and drug distance contributed to the predicted power which was the case for both data sources (Fig. 3d). The predictive power of our model of this logistic correlation model was analyzed using receiver operator curve (ROC) analysis, which showed an area under the curve (AUC) of 0.858 (Fig. 3e). The best-ranking drug combinations predicted by the model are provided in Supplementary Data 3d. When the model was applied to the independent DREAM benchmark data, the AUC was 0.735 showing that our model can be applied to external data (Fig. 3e). Synergies were relatively more often

observed in the literature and predicted for breast cancer and less often observed and predicted for lung tumors (both $P < 1 \times 10^{-5}$; Supplementary Fig. 3e).

Together, these analyses show a clear positive correlation between drug distance and the occurrence of synergy between the corresponding drugs. In addition, cell lines that show synergy are commonly relatively sensitive to these drugs, especially in cases where tumor-driver mutations are targeted. We were able to generate a prediction model for the occurrence of synergy which showed a good performance, also when applied to external DREAM data. We could validate our model in a drug-synergy screen showing a significant enrichment over the background. These data support our hypothesis and link our concept to personalized features.

**The drug atlas enables identification of multi-drug synergy in vitro.** Multi-drug (>2) combinations are difficult to identify given the enormous numbers of possible combinations of more than two drugs. For example: for 600 FDA approved cancer drugs there are $54 \times 10^{6}$ possible combinations of three drugs per cell line/patient. We noticed that multiple identified synergy pairs show connections on the drug atlas. In the cell lines U251, T98, and U87-MG, connected triangles can be observed where each pair of each axis has individually shown synergy between the drugs Torin1, Erlotinib, and Docetaxel (Fig. 4a).

Given that sensitivity and distance predicts synergy, we argued that dual synergies might predict synergies of multidrug combinations. We therefore experimentally validated a potential synergistic effect of Torin1, Erlotinib, and Docetaxel (Fig. 4b, all data are shown in Supplementary Data 4). We calculated the secondary synergy (see Methods), to make sure that each drug contributed to the synergy. We tested a panel of 21 cell lines for the putative synergy which resulted in a strong synergistic effect overall with combination indexes up to 0.18 (strong synergy shown by strong red color, cell line H4 and T98) leading to a loss of viability below 10% of the control (greyscale in Fig. 4b, lower panel of Supplementary Data 4). Glioma Sphere Cultures (GSC), i.e., primary cultures that faithfully resemble GBM tumors in their genetic and transcriptomic make-up, showed similar effects. The effective potency of each drug was increased 8–16-fold in the combination, and in some cases up to 64-fold (T98, U251). These data show that we have identified a synergistic multidrug combination where each drug enhances the effect of the other, leading to a strong synergy (average combination index of 0.46) with a severe loss of viability (average 92% reduction).

Since we argued that the effectiveness of dual synergy might be predictive for the effectiveness of multi-drug synergy, we therefore analyzed whether the magnitude of dual-therapy synergies correlated to the occurrence of multi-drug synergy. For all dual treatments, we correlated the corresponding dual combination index to the independently obtained multi-drug combination index. This clearly showed a significant correlation between dual synergies and the multi-drug synergy (Pearson correlation between 0.679 and 0.812, $P < 1.4 \times 10^{-3}$, Fig. 4c). Therefore, our methodology might be used to identify more multi-drug therapies based on dual-therapy effects.

Before we can apply the identified multidrug combination to a mouse model, it might be useful to focus on clinically relevant drugs for GBM patients, also taking toxicity and blood brain barrier transfer into account. We therefore chose to test a new panel of drugs that have overlapping targets with the previous set, but have better blood brain barrier crossing potential. We selected Osimertinib (Targrisso, AZD9291; targets EGFR), AZD2014 (MTOR1/2) and Docetaxel (Microtubules, molecular structures are shown in Supplementary Fig. 4a) which, based on literature

research, can reach concentrations in the brain that match effective in vitro conditions. When applied to a panel of cell lines/ primary cultures that previously showed a response to the previous set of drugs, this led to strong synergies in vitro as expected (Supplementary Fig. 4b). Together, by using our drug-atlas approach, we are able to identify a drug synergy between three drugs that would otherwise be difficult to achieve.

**Drug atlas identified combinations show synergy in vivo.** We analyzed whether our prediction model to predict synergy can be validated in relevant orthotopic mouse models. We selected the previously identified therapy of three drugs as well as the best-ranking drug combinations (see highlighted pairs in Supplementary Data 3d) for GBM, triple-negative breast cancer, melanoma, and leukemia models, and tested whether their respective predicted drugs showed synergy in vivo. No obvious toxicity was observed in these experiments (except for the triple combination, see below).

The combination of three drugs (Osimertinib, AZD2014, and Docetaxel) was tested in an U87-MG-FM (Fluc-Mcherry) orthotopic glioblastoma model, which showed a clear synergistic effect leading to 10–100 reduction of the tumor volume as measured from the luciferase levels (Fig. 5a, RLU average, combination index between 0.21 and 0.60) in vivo around days 14–18. The treatment resulted in a significant better survival ($P = 0.04$, Fig. 5f). All cross group significances are given in Supplementary Data 5. Some mice ($n = 3$ out of 7) experienced constipation due to toxicity of docetaxel with/without the other drugs (Fig. 5a). Progression occurred after day 14, either because the drugs were administered too shortly or because therapy resistance occurred.

We subsequently applied a predicted combination in GBM consisting of a combination of two drugs: the PI3K/MTOR/ microtubule inhibitor GNE-317 and docetaxel. This drug combination came out of the logistic multiple regression model as a top-ranking combination. GNE-317 has been shown to pass the blood brain barrier[31]. When these drugs were co-administered, a good synergy was observed (combination index between 0.56 and 0.80, Fig. 5b). The survival of the mice was significantly better ($P < 0.04$, Fig. 5f).

We also applied a predicted combination in the triple-negative breast cancer cell model MDA-MD-231-FM. We choose the BRAF inhibitor AZD628 in combination with the nucleoside analog Gemcitabine as top-ranking drug combination. After orthotopic transplantation and start of the treatment, a strong synergistic effect was seen (combination index between 0.08 and 0.11, Fig. 5c), resulting in a significant better survival ($P < 1 \times 10^{-4}$, Fig. 5f).

We then tested a predicted combination for the Melanoma model CHL1-FM. For this, we used the CDK4 inhibitor GCP-082996 and the nucleoside analog Gemcitabine, again the top-ranking drug combination. After orthotopic transplantation and treatment, a clear synergistic effect was seen (combination index between 0.62 and 0.68, Fig. 5d), resulting in a significant better survival ($P < 1 \times 10^{-4}$, Fig. 5f).

We finally tested a predicted combination for the Leukemia model BV-173-Gluc. Due to the metastatic nature of these experiments, the cancer cells were tagged with soluble Gluc which can be measured in the blood of the mice. We used the BCR-ABL inhibitor Imatinib in combination with the BCR-ABL inhibitor Dasatinib because this was the top-ranking drug combination. After orthotopic transplantation and treatment a synergistic effect was seen after 14 days (combination index <0.25, Fig. 5e), resulting in a significant better survival ($P < 1 \times 10^{-4}$, Fig. 5f). Synergy might, in this case, be driven by a

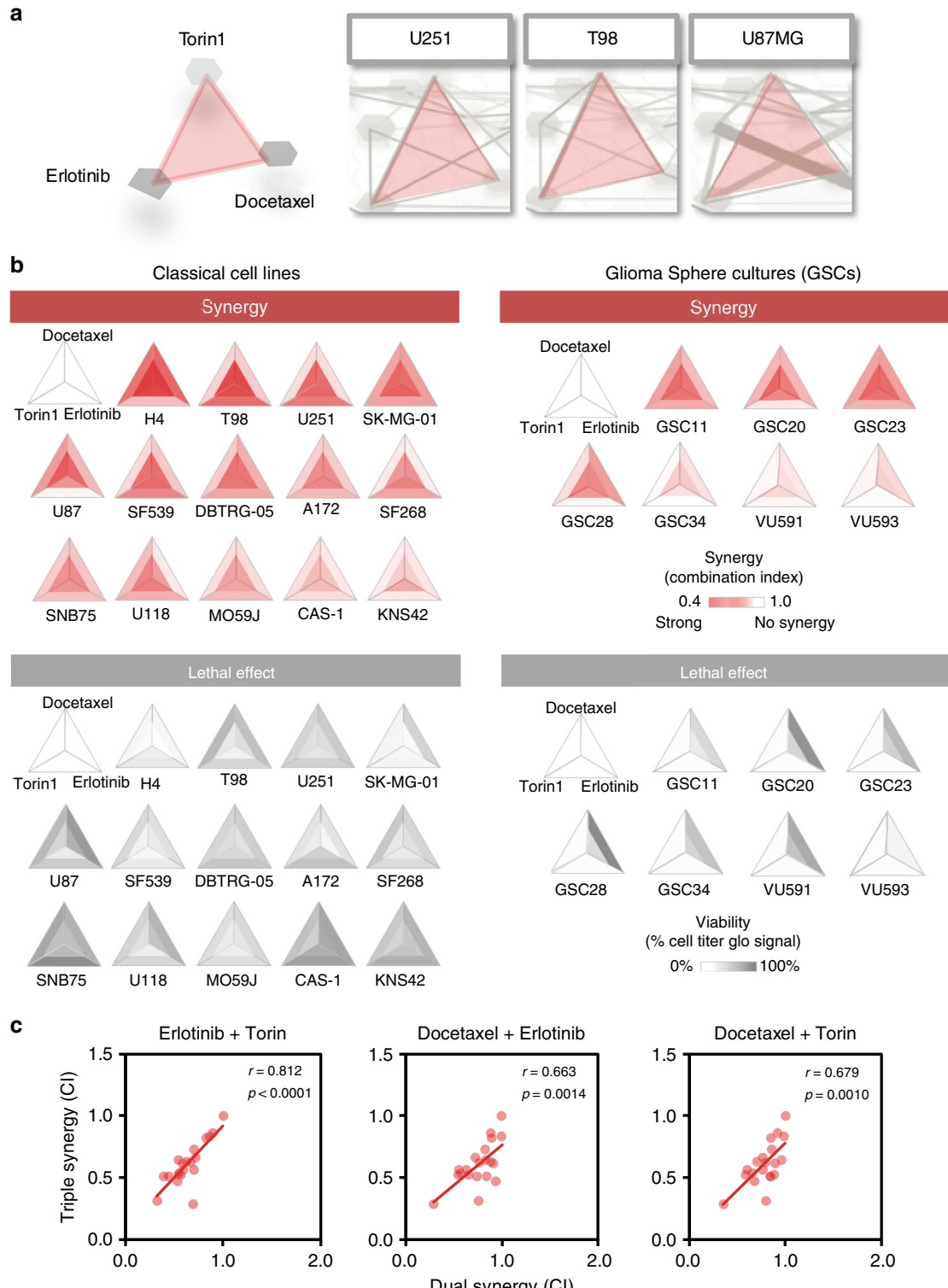

combination of optimal on-target (BCR-ABL) and off-target effects (Imatinib inhibits PDGFR and Dasatinib inhibits Src[32,33]) of these drugs resulting in complementing polypharmacology. One more model showed a significant survival effect as well as synergy of the combination (Supplementary Fig. 5a, b, MDA-MD-231 triple-negative breast cancer model) and two models showed a weak survival advantage without showing significance synergy (HT-29 colorectal cancer model and NCI-H460 non-small cell lung cancer model (see Supplementary Fig. 5c, d).

Together, the outcome of these in vivo experiments confirm the validity of our synergy prediction model in five independent mouse models and indicate that the prediction model has a translational value.

## Discussion

Selections of combinations of drugs that optimally match personalized features are pivotal for an efficient therapy. Since the number of possible combinations of drugs is enormous, we have used a rational approach to identify synergistic drugs rather than

**Fig. 4 Identification and validation of a synergistic therapy of three drugs. a** Plots showing a magnified part of drug atlas containing the dual synergy results. The plots enable to identify putative triple-synergistic drug combinations by connecting effective dual synergistic combinations in this case leading to identification of an Erlotinib, Torin1, and Docetaxel combination, which was validated in 21 cell lines. **b** Combination indexes of serial twofold dilutions of the three drugs when administered as a dual (outer triangle) or triple combinations (inner triangle). Both serum grown classical cell lines as well as serum-free cultured primary GBM cultures were analyzed. Synergy (shown in red) was calculated by the median effect principle[29] by calculating the added effect of the third drug on top of the effect of the first two drugs (secondary synergy, see Methods). For this, twofold dilutions that led to a IC50 effect were performed, using drug concentrations of Erlotinib (2–20 μM), Torin (0.4 μM), and Docetaxel (6.3–25 nM) as start concentrations. Lower panels in grey show relative viabilities after treatment with the three tested drugs. Drugs were diluted in a twofold manner and viability was assayed using CellTiter Glo 3D after 72 h. All data points were normalized to untreated controls. Experiments were performed in triplicate and repeated independently. Non-consistent results were repeated until consistent. **c** The combination indexes of measured dual synergies were significantly predictive for triple synergy as shown for 21 experimentally tested cell lines. r value is the Pearson correlation. *P*-value: Pearson correlation *P*-value.

a high-throughput drug-screen and biomarker-based approaches that are common practice in the field[12,17,24,34]. In particular, a long sought approach to discover multidrug ($n > 3$) synergies has so far been lacking in the field given the practical difficulties in experimental setup and capacity needed. Our drug atlas approach enables to identify these multidrug combinations which could increase therapy combination efficacy, reduce therapy resistance and could aid in designing optimal polypharmacological (i.e., multi-targeted) therapies. We used single drug dose–response data to construct the drug atlas. Since this drug atlas is based on normalized drug sensitivities, it can be seen as a quantifiable model of the relations between drugs. The validation of our hypothesis that unrelated processes might be important for occurrence of drug synergy came from curated synergistic interactions among 156 human cancer cell lines. These drugs were matched, when possible, to genetic mutations (i.e., onco-genic drivers). This showed that both drug distance and the drug sensitivity positively correlate to the occurrence of synergy. Independent validation of our model by in vitro drug screening of GBM cell lines confirmed our predictions. In addition, testing our model on the DREAM synergy benchmark dataset, additionally showed its value.

Both closely related as well as unrelated processes have been considered accountable for drug synergies: they result from intimate process connections (causing maximal target, pathway or feedback inhibition[35–39]) to less related parallel pathway connections that can cause synthetic lethal interactions[36,40–42]. Gayvert et al.[12] showed that synergistic combinations in mutant BRAF cell lines had a trend toward lower correlation of sensitivity over multiple cell lines, hence a drug-distance effect. Our method, that is based on common exclusive effects of drugs, is relevant for processes that are commonly only weakly connected. When these processes are simultaneously active in tumor cells, they offer a particular strong vulnerability given their independence. Thus, this provides a way to move from an already beneficial mutual exclusive action[13] to an even more beneficial synergistic mode of action. Our model therefore complements previous findings and concepts and provides a framework for understanding the relations of survival mechanisms.

Based on our distance model, we could generate a drug-synergy predictive model. As a proof of concept, we validated the synergistic effect of five drug combinations in vivo for GBM, triple-negative breast cancer, melanoma, and CML models in mice. The identified combination of three drugs also resulted in a synergistic response in vivo, resulting in a 10–100-fold reduction in tumor size in vivo. The other models showed a similar per-formance including an additional triple negative breast cancer model. Together with a lung and colorectal cancer model that did not show synergy, the success rate of synergy identification is six out of eight cases (75%, $P = 2.3 \times 10^{-9}$ over an estimated 8% background synergy).

A major obstacle for implementing combination treatment in the clinic is the occurrence of synergistic toxicities. In many cases, these toxicities are a result of additive toxicities because targets are shared between the combined drugs[37,43–45]. Although we noticed only minor toxicity during the in vivo experiments, the combination of three drugs led to toxicity, probably through epithelial damage of the colon. To enable to proceed with this therapy, a scheduling strategy might important to reduce toxicity without compromising the efficacy.

For both independent action as well as for synergistic inter-actions of drug combinations, therapy resistance might occur. This could be due to various compensation and independence mechanisms that can occur on a cellular level[46]. Drug independence and synergy do therefore not preclude therapy resistance but when optimally aligned, stronger and more lasting effects of drug combinations can be expected. In conclusion, by using single-drug dose–response data we could predict combination therapies and have found that independent (parallel) vulner-abilities represent an important class of drug combination targets. We have developed a method to identify these vulnerabilities which enabled us to predict multidrug combinations which could be validated in vivo with a high success rate. The atlas concept provides an important insight in how to predict effective com-bination therapies. Our method is scalable and forms a resource for future translational validation of our results.

## Methods

**Molecular features of the cell lines and drug targets**. Datasets used in this study are described in Supplementary Data 1 and refer to Sanger GDSC cancer cell lines (as well as Novartis/Broad CCLE data). For expression analysis, data of the Cosmic consortium were used (Gene expression analysis of 789 cancer cell lines using the Affymetrix HT- HG-U133A v2 platform, Source: EBI ID: E-MTAB-783[22]).

**Generation of the drug atlas**. Since drug dose–response datasets can be seen as collections of $n$-dimensional vectors, the similarity between these nonzero vectors (has at least one nonzero component) can be determined by calculating the cosine alpha of the inner product space, or in other words, the angle the drug-response vector for a specific drug has to all other drug-response vectors[47,48]. For all cal-culations, the relation between drug dose–response data over all cell lines was calculated as the cophenetic distance[49,50]. The cophenetic distance of two objects in a cluster tree is the depth of the branches separating both objects, and is defined by the following formula (Eq. (1)):

$$c = \frac{\sum_{i<j} (x(i,j) - \bar{x})(t(i,j) - \bar{t})}{\sqrt{\left[\sum_{i<j} (x(i,j) - \bar{x})^2\right]\left[\sum_{i<j} (t(i,j) - \bar{t})^2\right]}}, \qquad (1)$$

where C is the cophenetic correlation coefficient, which can be calculated from $x(i, j) = |Xi - Xj|$, the ordinary Euclidean distance between the ith and jth observations. $t(i, j)$ is the dendrogrammatic distance between the model points Ti and Tj. This distance is the height of the node at which these two points are first joined together. $\bar{x}$ is the average of the $x(i, j)$, and $\bar{t}$ is the average of the $t(i, j)$ The script is available at https://stat.ethz.ch/R-manual/R-devel/library/stats/html/cophenetic.html.

AUC values of the GDSC Encyclopedia (MGH data only[22], see also "Quality control: dataset quality" below) were stored in sorted data vectors as 1-AUC per cell line. The value of the AUC fluctuates between 0 and 1; zero reflecting the

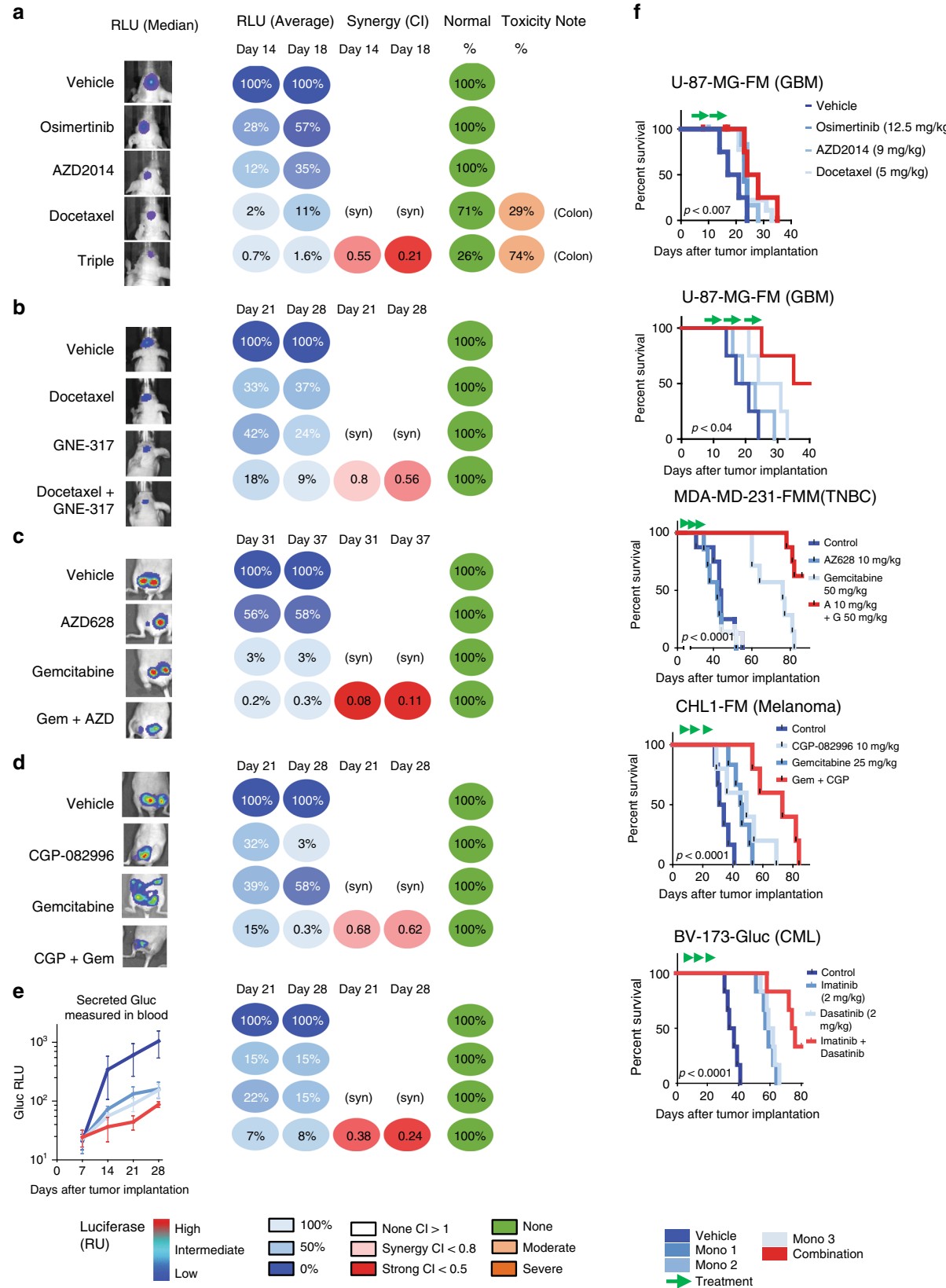

To generate the drug atlas Voronoi map, AUCs of GDSC-MGH dose–response experiments were converted into cosine alpha vectors. This distance matrix of all against all drugs was clustered using a hierarchical clustering algorithm with a Euclidean distance metric and pairwise average linkage resulting in a cluster tree separating the drugs based on their responses to drugs in different cell lines. This binary cluster tree was subsequently drawn as a flattened map using an adapted version of the Weighted Fast Voronoi Layout (WFVL) algorithm[24]. In a Voronoi

highest response. The distance of drug A with respect to drug B was calculated as 1 minus the similarity between the AUC data vectors. Dose–response data were clustered using Ward.D2 or average clustering. Within-pathway distances were calculated by using the ontology groups according My Cancer Genome, see Supplementary Data 1 for sources. Drugs with known cross reactivity over multiple pathways were excluded to calculate the within-pathway distance.

**Fig. 5 In vivo validation of predicted combination therapies. a** In vivo luminescence monitoring after orthotopic transplantation of Fluc-mCherry-tagged U87-GBM cells. Tumors were engrafted for 1 week and then treated with Osimertinib, AZD2014, and docetaxel (RLU median). Measurement of averages of luciferase activity (RLU average) is shown after 14 and 18 days showing a synergistic response (combination index between 0.55 and 0.21). Note that some toxicity was observed in docetaxel-treated mice. **b** Similar setup showing orthotopically transplanted Fluc-mCherry-tagged U87-GBM cells (median of each group is shown) after treatment with the PI3K/mTOR inhibitor GNE-317, and the microtubule inhibitor Docetaxel, resulting in a synergistic response (combination index between 0.56 and 0.80). **c** Measurement of luciferase activity in the triple-negative breast cancer cell model MDA-MD-231-FM showing the median after treatment with the BRAF inhibitor AZD628 in combination with the nucleoside analog Gemcitabine, resulting in a strong synergistic response (combination index between 0.06 and 0.11). **d** Measurement of luciferase activity in the orthotopically transplanted Melanoma model CHL1-FM showing the median after treatment with the CDK4 inhibitor GCP-082996 and the nucleoside analog Gemcitabine resulting in a synergistic response (combination index between 0.62 and 0.68). **e** In vivo luminescence measurement of tail vein-injected chronic myeloid leukemia (CML) BV-173-Gluc cells. Due to the metastatic nature of the transplantation, the cancer cells were tagged with soluble Gluc which can be measured in the blood of the mice, showing a synergistic decrease in luciferase activity when the combination therapy of the ABL inhibitor Imatinib together with the ABL inhibitor Dasatinib was applied (combination index between 0.24 and 0.79). **f** Kaplan–Meier curves showing a better survival of mice treated with the combination of drugs (see Supplementary Data 5). For all experiments, luciferase levels were normalized to levels of one week after injection. Toxicity monitoring consisted of assessment of body weight, hematopoietic-, liver-, and brain toxicity. *P*-value: *t* test (one-sided) of the median survival. The number of mice per group are shown in the figures.

---

diagram, a plane is divided in regions based on a set of sites on the plane. The borders between the regions are drawn where the distances between two sites are equal. This results in set of polygons. In a weighted Voronoi diagram, the distance to a site is calculated according to the weight of a site. A centroidal Voronoi diagram shifts sites to get an even distribution of sites over the plane by taking the aspect ratios of the polygon sides into account. By recursively applying a weighted centroidal Voronoi diagram to a cluster tree of data we can map this data structure onto a plane (see also Supplementary Fig. 1a).

Since this is a non-deterministic algorithm thresholds need to be set to finalize calculation. Furthermore, these calculations are computationally intensive, so heuristics are needed to compute this in acceptable time. Most important heuristic in the WFVL is the use of a "power diagram"; a transformation of the 2D plane to a 3D convex hull that enables fast calculation of the centroidals.

Additional heuristics that were used were developed in house and are described in the pseudocode below:

1. Calculate branch grouping threshold based on percentage clusters formed at tree depth d → branch threshold b
   Apply heuristics

2. Group if branch distance below threshold b
3. Identify early split of genes to be placed in corners based on similarity → grouped clustertree $T_g$

4. Recursively layout branches: Apply weighted Fast Voronoi Layout (FVL) per branch → approximate Voronoi map $V_a$

5. Smoothen $V_a$ map by re-applying FVL while maintaining relative positions → evenly distributed Voronoi Map V

Thickness of borders between regions reflects the cutoff in the cluster tree (the cophenetic distance); higher up results in a thicker border.

**Quality control: dataset quality**. Initially data of the combined pharmacogenomic encyclopedias of the Genomics of Drug Sensitivity in Cancer (Sanger GDSC[29]) consortia were used to generate the drug atlas, but the resulting clustering pattern showed that the difference between the WTSI and MGH drugs within this dataset, which had a more pronounced effect on the clustering than the actual dose–response data itself[23,51,52]. GDSC1000[27] also clustered independently from the WTSI and MGH data, probably because the coverage was much higher since all cell lines versus drugs were measured. However, within this data, similar as for the original GDSC (WTSI) data, there were stronger clustering differences observed than for the GDSC (MGH) data. Given this bias, we selected the GDSC (MGH) dataset which showed the most robust clustering (see Supplementary Fig. 1f).

**Literature synergy data curation: in silico identification and visualization**. In order to obtain control data for our approach, we retrieved drug-synergy data from the literature. A systematic literature search to identify all published synergy data for all Sanger GDSC (MGH) cancer cell lines was performed to gain insight into the usability of our drug atlas. Using GDSC (MGH) cell lines as a reference, we used Boolean operators to generate a full list all known synergistic drug combinations for these cell lines. For this, we used the following steps:

1. Pubmed was searched for <CCLE/GDSC cell line name> to check whether the cell line is annotated at all (wild cards <*> were used to identify alternative spelling of the cell line names).
2. If positive, pubmed was searched for < CCLE/GDSC cell line name > + <synerg*> (the wild card * ensures that both synergy and synergistic are found).

3. For each positive case, the PDF manuscript was checked whether the cell line (s) used was correct and which method was used to calculate drug–drug synergy. The following methods were allowed (of which the first two cover 92% of the cases):

 Combination Index Chou and Talalay
 isobologram analysis
 *t* test (in histogram or table)
 z-score>3
 Bliss or Loewe synergy
 Variants of the upper methods

Additive or antagonistic interactions were also recorded using the PubmedID.
4. In addition, Google scholar was searched for < CCLE/GDSC cell line name > + <synergy >" using wild cards as above. When more than ~100 hits were found, <methods to determine synergy (see point 3)> were added in the search term, and the provided text was manually checked for correctness.

Only peer reviewed papers annotated in Pubmed were considered. No ligand treatments were included. Papers mentioning synergy, but referring to biochemical or biophysical interactions were excluded. References for all found synergies are given in the Supplementary Referenes. Synergistic drug combination targets were annotated onto the atlas for each tumor type either as drug relation or target relation. Pairs of identified synergy pairs were visualized using Sankey diagrams (http://sankeymatic.com/build/).

**Synergy prediction model based on sensitivity, distance, and mutations**. The objective of the model is to predict drug synergy by using not only sensitivity data of different cell lines to individual drugs but also the drug atlas distance. Predictions with and without using the drug atlas distance were performed as to demonstrate its added value. For the data preparation, each row in the data corresponds to values for one cell line and a pair of drugs. As such, there is a lot of structure in the data, via the drug pairs. In addition to the distance between the two drugs from the drug atlas, the data include individual drug-sensitivity value as well as target information relating to each drug of the pair, where target information indicates whether or not the drug targets a gene known to be affected/mutated in the cell line at hand. Finally, it includes an indicator variable if the two drugs are known to display synergy. Per drug pair and cell line, the objective function first proposed (Eq. (2))

$$C\left[-s\bar{S} + dD_{12} + t\sum_{i=1}^{2} T_i\right],\tag{2}$$

where C is a binary variable indicating whether sensitivity information about the cell line is available for at least one drug $(C = 1)$ or not $(C = 0)$, S is the average of the sensitivities $S_1, S_2$ of the cell line for drugs 1 and 2, respectively, $D_{12}$ is the distance between the two drugs in the pair, and $T_1, T_2$ represent the average value of the targeted variable for drugs 1 and 2, relative to the cell line. Specifically, $T_i$ is equal to 1 if the drug targets a gene known to be affected (e.g., mutated) in the cell line, and 0.01 otherwise. In the above, $C, S_1, S_2, D_{12}, T_1, T_2$ are given, while s, d, t need to be estimated. We minimized this function for all cell lines relating to the same tissue.

**Optimization of the model**. The objective being to predict whether or not two drugs display synergy, we analyzed the logistic regression model with the synergy indicator as response when the covariates consisting of (1) the individual drug sensitivities, (2) their target information and (3) the drug distance, were taken along, per cell line. Note that rows where drug-synergy information is left as 0 may be unknown or no synergy. We know that the synergy variable $\theta_{ij}$ depends on the distance $d_{ij}$ between the drug pair $(i,j)$ on the drug atlas, and on the target information $t_{ijk}$, which indicates if at least one of the drugs targets a gene mutated in the

cell line $k$. We could write the synergy between the two drugs indexed by $i,j$ as (Eq. (3)):

$$\theta_{ijk} = \frac{t_{ijk}e^{d_{ij}}}{1 + t_{ijk}e^{d_{ij}}}, \qquad (3)$$

which yields a positive correlation between the synergy $\theta_{ijk}$ and the drug atlas distance $d_{ij}$ and that, if none of the drugs $i,j$ targets a mutation of cell line k, the synergy is close to zero. We can compute $\theta_{ijk}$ per pair $(i,j)$ for all available cell lines, then average them out. This simplistic approach does not involve a logistic regression. Alternatively, we can fit per tissue a logistic model (Eq. (4)):

$$\text{logit}(\theta_{ij}) = \alpha + \beta t_{ijk} + \delta d_{ij} + e_{ijk}, \qquad (4)$$

where now $\theta_{ij}$ indicates synergy between drugs $i$ and $j$ irrespective of the cell line, $e_{ijk}$ represents a normally distributed error with mean 0 and variance $\sigma^2$, and $i \neq j$.

The above does not involve the individual drug sensitivities. These could be included simply as covariates, as in (Eq. (5)):

$$\text{logit}(\theta_{ij}) = \alpha + \beta t_{ijk} + \delta d_{ij} + \gamma_{ik}S_{ik} + \gamma_{jk}S_{jk} + e_{ijk}. \qquad (5)$$

Note that the above do not make use of any model for the combined sensitivity, which is not observed in this case.

**Application**. The idea is to fit the model to the cell lines for which synergy is known. Note that there is no information if no synergy is also known. We will only use the cell lines for which sensitivity is available for both drugs. At this point, the other ones are not informative.

**Logistic regression**. The data contain an indicator variable Label that is 1 for cases where the two drugs display synergy, and 0 otherwise. Per tissue, we fit a logistic regression using all rows for which Label is 1, and then choosing at random the same number of cell lines (observations) for that tissue for which Label is 0.

**Testing covariates: drug targets**. We compared four model fits. Model IA relates only the target information (at least one drug targets the cell line modification) is (Eq. (6)):

$$\text{logit}(\theta_{ij}) = \alpha + \beta t_{ijk} + e_{ijk}. \qquad (6)$$

**Testing covariates drug targets and distance**. Model IB relates both target information and the drug atlas distance (Eq. (7)):

$$\text{logit}(\theta_{ij}) = \alpha + \beta t_{ijk} + \delta d_{ij} + e_{ijk}. \qquad (7)$$

This model can be used to assess the added value of the distance by comparing its results to those using model IA.

**Testing covariates drug targets and sensitivity**. The second pair of models involves the target information, as well as the individual sensitivities of the cell line to the drugs. Model IIA is (Eq. (8)):

$$\text{logit}(\theta_{ij}) = \alpha + \beta t_{ijk} + \gamma_{ik}S_{ik} + \gamma_{jk}S_{jk} + e_{ijk}. \qquad (8)$$

**Testing covariates drug targets, sensitivity, and distance**. Finally, model IIB is the same as model IIA, but also includes the drug atlas distance (Eq. (9)):

$$\text{logit}(\theta_{ij}) = \alpha + \beta t_{ijk} + \delta d_{ij} + \gamma_{ik}S_{ik} + \gamma_{jk}S_{jk} + e_{ijk}. \qquad (9)$$

This model can be used to assess the added value of the distance by comparing its results to those using model IIA.

**Correlations between drug sensitivity and synergy**. To correlate the occurrence of synergy with the drug sensitivity, we normalized the drug sensitivity per drug per tissue type to correct for the tissue specific dynamic range. For this, sensitivity per tissue type per drug was shifted to the normalized average value, which was set to 1 (average $^2$log IC50 concentration per tissue per drug). All sensitivities shown are the delta $^2$log IC50 compared to the normalized value.

**GBM cell lines and primary cultures**. The glioblastoma cell lines used for this study are listed in Supplementary Data 1. Cell lines were cultured in Dulbecco's Modified Eagle's Medium (Gibco™, Life Technologies) supplemented with penicillin/streptomycin (Gibco™, Life Technologies) and 10% fetal bovine serum (Gibco™, Life Technologies) and maintained at 37 °C with 5% CO₂ in a humidified environment. Cells were grown strictly in the log phase between and during experiments. Glioma Sphere Cultures (GSCs) were obtained from single patient surgical specimens at MD Anderson (procedure is described in Bhat et al.[53]) or at the Vrije Universiteit medical center (VUmc) Amsterdam. At the VUMC, tumors were washed twice with phosphate-buffered saline (PBS) in a Petri dish. The tumor was cut into small pieces and treated with Accutase, containing 5% sterile

filtered EDTA (0.5 M, [pH 8]) and 4.5% DNAse I (10 mg/ml diluted in Hanks' balanced salt solution (HBSS), Roche Life Science). The tumor material was incubated for 30 min at 37 °C, and dissociated every 10 min by pipetting up and down. Next, the tumor material was passed through a 100-µm cell strainer to obtain a near-single-cell suspension. The suspension was centrifuged for 5 min at 1000 rpm, and supernatant was discarded. To lyse erythrocytes, 1 ml of E-lysis buffer was added and incubated for 5 min at 37 °C, followed by centrifugation for 5 min at 1000 rpm to remove the supernatant. Glioma Sphere Culture (GSC) cell lines were grown in Neurobasal-A Medium (NBM; supplemented with 1× N-2 supplement, 1× B-27 supplement, 0.1% heparin, 20 ng/ml EGF, and 20 ng/ml bFGF (Peprotech) and primocin (Gibco). The medium was refreshed at least once a week including 20–50% old medium, depending on the growth rate of the GSCs. Spheroids were dissociated with Accutase (PBS containing 0.5 mM EDTA·4Na and 3 mg/L Phenol Red, Sigma-Aldrich) and kept up to a maximal size to prevent occurrence of a necrotic core. Spheres were centrifuged for 5 min at 200×g, and the supernatant was discarded. Spheroids were resuspended in Accutase and incubated for 3–5 min at 37 °C, again centrifuged for 5 min at 200×g and afterwards plated in a flask.

**Drug screens**. Drug screens were performed using the following drugs of which the drug itself or its target was previously shown to be active against glioblastoma: Gemcitabine[54] (Selleckchem), Rapamycin[55] (Sigma-Aldrich), Docetaxel[56] (Selleckchem), Erlotinib[57] (Selleckchem), JNK inhibitor VIII[58] (EMD Millipore), Akt inhibitor VIII[59,60] (Sigma-Aldrich), Crizotinib[61] (LC laboratories), Torin1[62] (LC laboratories), Pac-1[63] (Sigma-Aldrich), Embelin[59] (Sigma-Aldrich), AZD6482[58] (Tocris Bioscience), AS601245[64] (EMD Millipore). Drugs were dissolved in DMSO.

**Screen optimization**. Plating density of cell lines was determined to obtain ~90% confluency after 96 h. Cells were counted with Coulter Counter (Beckman Coulter), and cells were seeded in hexplo in 96-wells plate with a density of 5000, 4000, 3000, 2000, or 1000 cells per well in 200 µl DMEM complete medium (+10% FBS and 1% P/S). Cell viability was determined 96 h post seeding using CellTiter-Glo® 3D (Promega) viability assay (which uses ATP quantification as a readout for metabolically active cells). In total, 150 µl medium was removed from the wells, where after 50 µl CellTiter-Glo® 3D was added. Luminescence was captured at OD1, 400-ms integration time from Greiner 96 Flat Bottom White Polystyrene (GRE96fw_chimney) plates. After 20–30 min of incubation, the total lysates were transferred to white polystyrene 96-wells plates (Greiner), and relative light units were determined by Infinite® 200 Microplate Reader with a CONNECT plate stacker system (Tecan). Cell viability for these optimizations was confirmed by the amount of attached growing cells and by Crystal Violet staining. Cells were seeded as described previously and 96 h post seeding all medium was removed by pipetting. The cells were fixated with 100 µl 3.7% formaldehyde in PBS for 20 min and stained with 100 µl Crystal Violet solution (0.1% Crystal Violet, dissolved in 25% Methanol) for 30 min. Subsequently, the Crystal Violet was discarded, and the plates were rinsed with demi water and tapped onto a dry tissue, until no dye appeared on the tissue. Plates were air dried at room temperature for at least 4 h. Crystal Violet staining was dissolved in 100 µl of 1% SDS in demi water, and after 5 min at shaking plate the OD was determined in Tecan reader at 540 nm. Experiments were performed in triplicate, and were independently confirmed in replicate experiments. Toxicity of the chosen drug pairs was taken into account by using normal human astrocytes, fibroblasts, and neural stem cells.

For plating of glioma organoid/sphere cultures, organoid/spheroids were dissociated into single cells using Accutase as described above. Cell counting was performed by using a coulter counter (Beckman, US). Per cell line, two rows (per row six wells) of respectively 5.000 and 2.500 single cells diluted in 100 µl NBM were plated in non-coated round-bottom 96-well plates, in triplicate. Formation of multicellular organoids took 4 days; consequently day four was indicated as "Day 0". For optimizations, every 2 days, growth was measured by determining the spheroid volume (mm³) and by measuring the viability using CellTiter-Glo® 3D as above. Cells were incubated for 30 min with CellTiter-Glo® 3D, and subsequently luminescence was measured. Measurements were averaged, and ratios were calculated by day X/day 0. Viability was measured on days 0, 2, and 4. To determine drug toxicity for primary cultures from the VU Medical Center, organoid cultures were generated using non-coated round-bottom 96-well plates in a 6 × 9 matrix. One row was used as control, whereas the other eight rows were used for drug titration in triplo (8 × 3). Per well, 3000 single cells in 100 µl NBM were plated. After organoid formation, drugs were added. Drugs were diluted 1:2 in NBM per row. Three days after addition of drugs cell viability was determined using CellTiter-Glo® 3D, using the same procedure as described above. For analysis, the average cell viability of the controls and the different drug concentrations was determined. Next, the cell viability in percentages was calculated (average cell viability drug concentration X/average cell viability control*100). IC50 concentrations were determined using GraphPad software, plotting nonlinear regression curves and defining the absolute IC50 concentration or area under the curve (AUC) surface.

**IC50/AUC determinations**. Inhibitory concentrations leading to 50% viability (IC50) were determined before synergy testing was performed. Optimal seeding densities and drug concentrations were determined by titration in 100 μl DMEM complete medium according to a 6 × 8 format. Twenty-four hours after seeding, 100 μl of the diluted drug was added. Drugs were titrated from a high concentration down in a nine step, threefold dilution series (up to 20,000-fold dilution). IC50 (and area under the curve [AUC]) concentrations were determined using Graphpad Prism software.

**Drug–drug-synergy determination**. Titrations to determine drug synergy were done by plating cells in optimal densities in a 6 × 7 format in 96-wells plate in 100 μl DMEM complete medium. After 24 h, cells received treatments of 50 μl of drug 1 and 50 μl of drug 2, which were serial diluted twofold diluted in each step, resulting in a non-treated control and an additional five concentrations for each drug, together forming a matrix of 36 different concentration ratios for both drugs. For drug-synergy determination, drugs were titrated in a window where the IC50 concentration was chosen as the highest concentration to prevent stochastic (low cell density) and off-target artifacts. Absolute IC50s were used, except in the case of Rapamycin where the relative IC50 was used (absolute IC50 was never reached). For AZD6482 and JNK inhibitor VIII, it was in some cases not possible to determine the IC50. The negative control, that received no drug, was replicated in sixfold to obtain a more balanced design of untreated versus treated wells. Assays were performed at least in triplicate in 96-wells plates (Nunc) for classical GBM cell lines or in low-adhesion plates when primary cultures were used. Three days after, drug treatment cell viability was measured using CellTiter-Glo®. Per 96-wells plate the average cell viability of each drug condition in percentages was calculated using DMSO-treated cells as controls. After data normalization, synergy calculations were performed (example shown in Fig. 2i).

**Multi-drug combinations**. Drugs used for the multi-drug experiment were Erlotinib, Torin1, and Docetaxel. Cells were plated as described for dual-drug combinations and 24 h post seeding 50 μl of drug 1 and 2 were added in a five times twofold dilution series in two directions. In total, 18 × 96-wells plates were necessary for one multi-drug experiment: 50 μl of third drug was added per set of three plates in one concentration over the whole 6 × 7 format. In total, five concentrations of third drug were added, and DMSO-treated cells served as control. For primary cultures, four round-bottom 96-well plates were used, 3.000 cells per well were plated in a 6 × 9 matrix. After 4 days, drugs were added to the organoids/spheroids. Per drug, three different concentrations were used (diluted 1:4 for, respectively, Erlotinib and Torin1, 1:2 for Docetaxel). After 72 h of treatment, 200 μl medium was removed from the wells, after which 50 μl CellTiter-Glo® 3D was added. Read-out methods and data calculations were performed as described above.

**Synergy calculation**. Synergy was determined using the median effect principle by Chou and Tallalay[29]. The combination index of an amount of n drugs was be calculated by using an n-dimensional adapted formula, Eq. (10)).

$$CI_{(n \text{ drugs})} = \frac{\sum_{k=0}^{n} \left(\frac{1}{V_n}\right) - \left(\frac{n-1}{100}\right)}{\left(\frac{1}{V_{1..n}}\right)}, \quad (10)$$

where $V_n$ is the normalized measured viability (in %) of each drug separately for n drugs. $V_{1..n}$ is the actually measured viability (in %) of the combination of 1..n drugs. A combination index lower than 0.8 is indicated as drug synergy, a combination index of 1 is indicated as additive and a value higher than 1 is indicated as drug antagonism. For each cell line, the combination index (measured synergistic effect, % cell viability) was determined and subsequently the theoretical additive effect (% cell viability)—the effect of the drug combination when the drugs in theory would not have worked synergistically—was calculated to compare the measured synergistic effect with the theoretical additive effect. Loewe[65], Bliss[66], and HSA (highest single agent) calculations of synergy were performed using Combenefit[67], software is available at the following website: https://sourceforge.net/projects/combenefit/). Thresholds for synergy were determined by linear interpolation using the combination index of 0.8 resulting in the threshold for Loewe synergy at 7.8 and for Bliss additivity at 2.5.

**Multi-drug experiments**. For $n = 3$ drugs, synergy can be expressed either as the sum of all three drugs (primary CI) as well as the effect of the n (third) drug on top of n-1 (two) other drug combination; i.e., [DA+ DB] + [DC], [DB + DC] + [DA], and [DA+ DC] + [DB] (secondary CI)[68]. The primary CI was used for all double-synergy experiments and the in vivo experiments. The secondary CI was used for all in vitro multi-drug experiments. To calculate the secondary combination index, the combined effect of the first two drugs is taken as a single-drug effect and the synergistic effect of the third drug is calculated on top of this

combination (Eq. (11)).

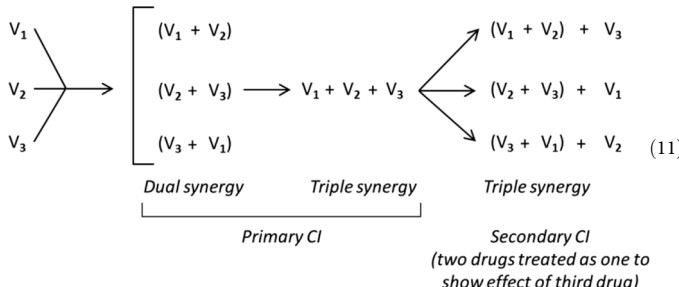

(11)

**Interassay reproducibility**. For some synergy determinations, inter-experimental reproducibility in the dose responses was observed. To solve this, the cultures were strictly cultured under log-expanding conditions between and during experiments. Experiments were repeated which resulted in proper window where the maximal dose led to a 50% viability readout.

**Statistics**. All Statistical analyses were performed using Graphpad Prism 5 software, except distance models (performed in R). The survival distributions were estimated using Kaplan–Meier methodology using the log-rank test. To enable to calculate the correlations between multiple drug sensitivities and the combination index, the relative sensitivity for each drug was normalized over all tested cell lines ($n = 20$) and expressed as 0 (most sensitive) to 1 (least sensitive). This enabled to calculate the average sensitivity for two drugs for each cell line and to correlate these values to the CIs.

**Ethical approval and mouse housing**. Studies were performed in accordance with the European Community Council Directive (2010/63/EU) for laboratory animal care and the Dutch Law on animal experimentation and when performed in a facility at Massachusetts General Hospital accredited by the Association for the Assessment and Accreditation of Laboratory Animal Care (AAALAC). Studies were approved by the Animal Welfare Body (IVD) of the VU and VUMC (in Amsterdam) and Institutional Animal Care and Use Committee (IACUC) in Boston. All experiments meet ARRIVE guidelines[69]. Four to 6-week-old female Athymic Nude-Foxn1nu mice were purchased from Harlan/Envigo, and used after 1 week of acclimatization. All animals were housed in one cage and kept under filter top conditions, receiving ab libitum water and food.

**In vivo efficacy testing**. For the GBM model, U87-MG cells were orthotopically injected into 8-week-old female Athymic Nude-Foxn1nu mice. Mice were anesthetized using isoflurane. The analgesic Temgesic was used at 0.1 mg/kg. In all, 0.3 mg/ml stock (Reckit Benckiser) was diluted 15×, and 50 μl was used per 10 g s.c. In addition, paracetamol (Bayer 120 mg/5 ml) was added 8× diluted to drinking water 1 day before the procedure. Lidocaine (VUMC Apotheek 13G15-001A, 1000 mg per 50 ml) was added to the skin surface. FM (Fluc-mCherry) tagged U87-GBM cells ($n = 400,000$) were injected using 5 μl PBS. Cells were injected using a stereotact injection device and injected in the striatum (0 = bregma) lr −2, tn 0.5, tb −3 mm. Injection speed was 2 μl per minute, followed by a 2 min lag time. One week after tumor engraftment, mice were treated as described under the sub-acute In vivo toxicity testing section. Luciferase activity was measured twice a week as described before[70]. In case the mice had to be taken out of the experiment due to tumor growth and/or weight loss, the last observation carried forward method was used to compensate for the loss of information in each group (defined at randomization). Progressive disease is defined as the last time point before disease progression (i.e., weight loss).

For the GBM models, mice were divided into four groups and treated with: vehicle (1% DMSO), Docetaxel ([71]; 5 mg/kg) and GNE-317 (40 mg/kg), and Docetaxel in combination with GNE-317. Drugs were administered intraperitoneally 4 days per week for 3 weeks. In the multidrug experiment, the same conditions were used and for Osimertinib[72], 12.5 mg/kg was given orally for two weeks together with AZD2014[73] at 9 mg/kg, given intraperitoneally. Docetaxel (5 mg/kg) was given intraperitoneally every other day, i.e., at days 1, 3, and 5 of each of the 2 weeks. Drugs were given individually or in combination.

For the triple-negative breast cancer model, MDA-MB 231 cells ($5 \times 10^6$) were injected into the fat pad of 8-week-old female athymic mice. Briefly, animals were restrained at upright position, and the needle was gently insert into the 4th mammary fat pad proximal to the nipple, bevel up, and 2–4 mm under the skin. When tumors were palpable, mice were divided into four groups, and treated with: vehicle (1% DMSO), AZ628 10 mg/kg, Gemcitabine 50 mg/kg, and AZ628 in combination with Gemcitabine. Drugs were administered intraperitoneally 4 days per week for 3 weeks.

For the melanoma model, CHL-1 ($5 \times 10^6$) cells were injected intradermally into the rear flanks of 8-week-old female athymic mice in 50 μl of PBS mixed with 50 μl Matrigel (Corning, NY). When tumors were palpable, mice were divided into four groups and treated with: vehicle (1% DMSO), CGP-082996 10 mg/kg,

Gemcitabine 25 mg/kg, and CGP-082996 in combination with Gemcitabine. Drugs were administered intraperitoneally 4 days per week for 3 weeks.

Bioluminescence imaging was performed using the Xenogen IVIS 200 Imaging System (PerkinElmer, Waltham, MA). The system is composed of an imaging chamber, gas anesthesia system which is connected to an oxygen cylinder and isoflurane tank, and a highly sensitive cryogenically cooled charge-coupled device camera. Fresh luciferin solution is prepared by dissolving D-luciferin powder (Gold Biotech, St. Louis, MO) in PBS at 25 mg/mL. Mice were injected intraperitoneally with 150 mg/kg body weight of luciferin and transferred into the image chamber. Imaging was acquired 10 min post-luciferin injection, and the image intensity was quantitated using the Living Image software 4.3.1 from PerkinElmer. Measurements of tumor size were also taken every 3 days using digital calipers, and tumor volume was determined by the following formula: volume = (length × width × height) × 0.52.

For Leukemia model, 8-week-old nude mice were sub-lethally irradiated with 120 cGy 24 h before the intravenous (i.v.) injection of $3 \times 10^6$ BV-173 through the tail vein. Mice were treated with: vehicle (1% DMSO), Imatinib 2 mg/kg, Dasatinib 2 mg/kg, and Imatinib in combination with Dasatinib. Drugs were administered intravenously 4 days per week for 3 weeks. At indicated time points, 5 μL of blood was withdrawn using a pipette Imaginib tip from a small incision at the tail tip of conscious mice and immediately mixed with 1 μl of 20 mM EDTA. Gluc activity was then measured using a plate luminometer (BioTek instruments, Vinouski, VT) after injecting 100 μL of 100 μM coelenterazine and acquiring signal over 10 s.

**Histopathological, hematological, and liver toxicity analysis**. After completion of the in vivo experiment (day 10), the brain, colon, and livers were fixed in 4% formaldehyde/PBS for 24 h, dehydrated with alcohol, embedded in paraffin, and tissues were then sliced into 4-μm-thick sections. Hematoxylin–eosin (H&E) staining of the sections and histopathological analyses were performed by Prof. Pieter Wesseling (department of Pathology, VUMC). Blood samples were collected in nonheparinized EDTA-coated Eppendorf tubes, and complete blood counts were determined with a COULTER® Ac·T diff™ Analyzer (Beckman Coulter, Miami, FL, USA). Furthermore, blood smears were prepared and stained using a May-Grünwald-Giesma protocol. Staining and the differential blood count (% of each type of white blood cell) were performed by the VUMC, department of hematology. The hematological parameters assessed were: hemoglobin concentration (HB), red blood cells (RBC), white blood cells WBC, and differential leukocytes (neutrophils, lymphocytes, and monocytes). In case of suspected relavance, liver toxicity was determined by measuring the liver enzymes Alanine aminotransferase (ALAT) and Aspartate Aminotransferase (ASAT) in plasma using an IFCC assay on the COBAS 8000 (Department of Clinical Chemistry, VU Medical Center). Vendor reference values were obtained from Envigo. After the animals were sacrificed, organ damages were analyzed. In the present studies, mice tolerated the treatment with no significant toxicity, except for the therapy of three drugs. We did not observe significant difference among the control and treatment groups at the evaluated dose/time point.

**Reporting summary**. Further information on research design is available in the Nature Research Reporting Summary linked to this article.

## Data availability

All public sources used for the project are provided in Supplementary Data 1. All other relevant data that support the results of this study are available from the corresponding author upon reasonable request.

## Code availability

Scripts are available through Github (https://github.com/bartwesterman/drug-atlas) together with a demo that runs the code/software in example data (NatComm model.csv) and typical run time. The script to create the Vonoroi diagrams is available upon reasonable request.

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

## Acknowledgements

This project was supported by the Dutch Brain Cancer Society (Stichting StopHersentumoren) grants 2014-006 and 2015-009, the Dutch Cancer Society grants KWF-4874 and KWF-11038, Brain Tumour Charity Grant 488097 and APCA-PoC-2017. We remember first co-author Piet Molenaar who sadly passed away during the final stages of publishing this collaborative project. Sjors van Heuveln is kindly thanked for providing the multi-drug identification algorithm. Beheshta Haydari and Louise de Vos Klootwijk are kindly thanked for their contributions to the data curation effort and the in vivo validations. Requests regarding applications of the drug atlas can be sent to B.A.W.

## Author contributions

B.A.W., J.K., and P.M. developed the conceptual framework. R.S.N., J.T., F.M.G.C., I.R., R.D., T.L., Y.B., N.P., J.A.M.S., E.B., M.C.L., and K.L. conducted all laboratory experiments. P.M., R.M., S.V., P.K., W.W., J.K., and B.A.W. performed the bioinformatics analyses. B.A.W., J.K., P.S., T.W., D.N., R. Verhaak, and R. Versteeg, F.F.L., E.S., B.G.B., L.J.A.S., L.V., C.W., B.A.T., D.B., B.J.S., and B.T. enabled the project by providing resources and supervision. B.A.W., R.S.N., J.K., P.M., and J.T. wrote the paper. F.M.G.C. and I.R. contributed equally. All authors reviewed the paper.

## Competing interests

The authors declare no competing interests.
