## [Peer Review File · Nature Communications]

Reviewers' comments:

Reviewer #1 (Remarks to the Author):

The core concept behind the proposed method is promising. Basically that the best synergies will occur between drugs targeting very different mechanisms, as revealed by how they affect many cell lines. This idea is not completely novel, especially because most cancer combination regimens specifically focus on diverse mechanisms, in order to overcome drug resistance, but the drug response atlas does neatly capture a way to identify useful combinations.

However, the execution of this idea and the validation experiments are flawed. Moreover, the authors conflate the synergy observed in an cell line combination experiment with the increased durability of effect seen in their animal experiments. The result is that the logical connections between drug-response -> synergy -> in-vivo response are just not convincing.

Detailed points:

- * The drug atlas generation is a nice way of comparing drug response profiles across all the cell lines. This is a novel approach that very nicely organizes drug pairs in a way that will be informative.
- * The literature synergy data generation is very unconvincing. Searching for CellName+"synerg*"+"Combination Index" will provide very misleading results. Many cell lines are known by differing descriptors, and the authors haven't described how they addressed that problem. Synergy as a term could relate to what was found or what was not found.
- * Also searching for Combination Index will only find synergies using the Chou+Talalay approach, so it's unclear how the authors came up with combinations listed as Bliss synergies.
- * Moreover, how the authors set their cutoff of "~100 references" for a combination, and curated their hits is not described. I'm guessing that much of the curation was manual inspection of many manuscripts, which probably explains why so few "synergy" calls (244) are determined. Some combinations known to synergize in these cell lines do not appear to be in the list. It would be very helpful if the supplementary table S1B-Curated-Data put the reference alongside each hit, so that it could be checked more easily.
- * A much more convincing analysis would have looked at the recent DREAM challenge results, and worked directly with observed synergy data rather than the literature synergy determination above.
- * For the validation experiments, the choice of Chou&Talalay method for synergy determination is unnecessarily complex, and using the much simpler Bliss or HSA models would have found much the same results.
- * The authors conflate the synergy observed in a cell line combination experiment with the increased durability of effect seen in their animal experiments. Those are two very different things - one relates to mechanistic interactions between targets in a cell, and the other relates to resistance that develops in a heterogeneous xenograft tumor. In fact, many combinations used in the clinic because they extend durable responses are not synergistic at all in the sense of a combination index. Indeed, a recent experiment (Palmer & Sorger 2017 Cell) showed that anti-resistance combinations rarely show any synergy whatsoever.
- * The Discussion makes statements that are not well informed. The claim that drug synergies were uniformly considered to arise from "intimate process connections", overlooks that many in the

literature have long assumed that distant targets would synergize better. In fact, synergies are known to happen in both situations, with different kinds of synergy: synthetic-lethal-like for parallel and potentiation-like for serial targets (Lehar et al. 2007 Mol Sys Bio). The synergy between Dasatinib and Imatinib is not at all surprising when you consider that they both have multiple targets (in the case of Dasatinib, many with comparable potency), which basically ensures that there are many targets involved in the interaction.

* The manuscript reads like a rough draft. There are repeated statements, miss-spellings, and inaccurate figure references (where what's described doesn't match the figure). The color scales on some figures aren't even consistent within one panel (see, eg. Fig.2G-viability).

* There are many missing citations which are relevant to this work, both in terms of past combination screens that have been run, and methodologies used.

Reviewer #2 (Remarks to the Author):

Narayan et al showed a novel perspective on how to study drug sensitivity to inform the selection of drug combinations with synergistic effects in cancer treatment. The authors used publicly available data from various sources (2012 - 2018) to develop the computational approach and validate its predictions. The major claims of the paper are i) the drug distance obtained using the proposed drug atlas, together with drug sensitivity can be predictive of drug synergy; ii) the dual synergy can be utilized to predict the effectiveness of the synergy in triple drug combinations; iii) triple and dual drug combinations that showed synergy predicted by the proposed approach were validated in vivo.

There are various approaches to predict the synergy in pairwise drug combination screens however the multi-drug ($N > 3$) combination synergy estimation is still lacking. This fact makes the paper of interest for cancer drug discovery. Another strength of the study is the experimental validation of the top ranked drug combinations in vivo. Overall, the proposed strategy is not complicated, which can allow for a simple extension in the lab or the clinic. The manuscript sections are well written and easy to follow, pictures are of good quality and readable. While the approach has merit, I think the authors have to improve some elements of the manuscript to increase its potential impact.

First of all, the manuscript has a poor description of the methods and techniques that authors applied for their study. The information from Supplemental Methods is not sufficient for making their work reproducible, and thus weaken the opportunity for other scientists to utilize the proposed strategy. Second, there is a lack of formalization when it comes to the drug/target distances on the drug atlas. The authors used the cophenetic distance to show that the distance of synergistic drug pairs significantly exceeds the average distance between the drugs. However, when it comes to concrete dual or triple drug combinations that are used for validation, the authors refer to 'high distance drugs' without specifying the exact distances or any threshold they used to filter out small distance drugs. This information has to be clearly stated in the paper to avoid subjective drug/target distance evaluation. Finally, the weakest point of the paper is the proposed synergy prediction model, as there is a poor description of the model, no information present about accuracy evaluation in the model training and no data available to check the rankings of the drug combinations based on the model.

Please, find more specific comments below.

Major comments:

1. Given that only small molecule drugs were used in the studies, the authors should state

whether the proposed approach is limited to those or justify that it can be extended for other drugs (like immunotherapeutics and perhaps antibody drugs).

2. The authors collected drug response data from GDSC and CCLE. However, there were concerns about the reproducibility of the drug screen studies (see e.g. PMID 24284626). The authors are advised to confirm that these multiple data sources are in good consistency.

3. In the Supplemental Methods the authors describe the way how they generated the drug atlas. However, it can be done in a better way, so that a reader that is not familiar with the listed methods can understand the main logic behind them. Thus, the description for such techniques as calculating branch grouping threshold, weighted FVL and map smoothing has to be improved.

4. It is not clear how synergistic interactions were mapped to the drug atlas and thus visualized in figures S1E and S1F.

5. In order to obtain ground-truth data for the approach, the authors performed literature survey. They included papers that fulfilled their searching criteria. However, there is no information provided what thresholds were applied on different synergy metrics, whether those were consistent within the papers, were there cases of overlapping synergistic drug combinations in papers (how the authors evaluate synergy in this case), were there cases where drug combinations showed controversial scores within different papers.

6. In the drug-drug synergy experiments the authors mention that per drug the start concentration was determined based on the IC50 curves. It will be useful to know how exactly the concentrations were chosen for the experiments: whether they had to span uniformly around the IC50 values or the starting concentrations were close to IC50 only. It is also stated in the paper that only 3 concentrations were utilized in the drug-drug experiments. Is this sufficient to capture the whole scope of the treatment response? This question again refers to the rational of the choice of the concentrations that was not clearly stated in the paper.

7. The authors used the median effect principle by Chou and Tallalay to calculate CI. Have they tried other metrics for comparison (e.g. Bliss mode and ZIP model, PMID 26949479). It will be also interesting to see the quantitative difference between the primary CI and secondary CI.

8. It is unclear how the multiple linear regression model training was performed (how did they estimate the accuracy of the model?) as well as how exactly the weights s , d , t , c were chosen. The log-rank test for prioritizing drug combinations has to be also explained. The authors can provide an xlsx file containing the rankings for the combinations.

9. The authors refer to 'high distance drugs' when performing experimental validation. A formal definition should be created to be able to separate high and small distance drugs. It might be helpful to see the table or a plot where sensitivity/synergy of drug combinations is compared to the corresponding drug distances from the drug atlas.

Minor comments:

1. In the Figure 3C description, there is (D) character instead of (C). The description does not correspond to the actual Figure either (it is a correlation plot, what is r value ?)

2. Line 256 use 'too' instead of 'to'.

3. Line 243 in Supplemental Methods - remove one extra 'distance'

We would like to thank the reviewers for their excellent input and for the time spend. We have changed the manuscript according to this input and feel that in the manuscript is now markedly improved. Please see our point-by-point responses below.

Reviewer #1 (Remarks to the Author):

General remark 1: The core concept behind the proposed method is promising. Basically that the best synergies will occur between drugs targeting very different mechanisms, as revealed by how they affect many cell lines. This idea is not completely novel, especially because most cancer combination regimens specifically focus on diverse mechanisms, in order to overcome drug resistance, but the drug response atlas does neatly capture a way to identify useful combinations.

Response: We appreciate these positive comments of the reviewer.

General statement 2: However, the execution of this idea and the validation experiments are flawed. Moreover, the authors conflate the synergy observed in an cell line combination experiment with the increased durability of effect seen in their animal experiments. The result is that the logical connections between drug-response -> synergy -> in-vivo response are just not convincing.

Response: Due to this critical note we now removed the implicated link between in vitro synergy and durability of the response. The experimental results do not (sufficiently) support this and distract from the relevant observation that synergy is actually measured in vivo. A clearer explanation and provisions are now added to the manuscript. Accordingly we changed this throughout the manuscript.

Detailed point 1: The drug atlas generation is a nice way of comparing drug response profiles across all the cell lines. This is a novel approach that very nicely organizes drug pairs in a way that will be informative.

Response: We appreciate this positive remark.

Detailed point 2: The literature synergy data generation is very unconvincing. Searching for CellName+"synerg*"+"Combination Index" will provide very misleading results. Many cell lines are known by differing descriptors, and the authors haven't described how they addressed that problem. Synergy as a term could relate to what was found or what was not found.

Response: Thank you for this critical note. The given description in the materials and methods was too brief. We have now provided the full methodology that was used to identify the references which is now correctly described in the supplemental methods and supplemental Table S1B (Curated data), which shows which method that was used to determine the synergy for each case.

Detailed point 3: Also searching for Combination Index will only find synergies using the Chou+Talalay approach, so it's unclear how the authors came up with combinations listed as Bliss synergies.

Response: This points is now corrected, see the previous point.

Detailed point 4: Moreover, how the authors set their cutoff of "~100 references" for a combination, and curated their hits is not described. I'm guessing that much of the curation was manual inspection of many manuscripts, which probably explains why so few "synergy" calls (244) are determined. Some combinations known to synergize in these cell lines do not appear to be in the list. It would be very helpful if the supplementary table S1B-Curated-Data put the reference alongside each hit, so that it could be checked more easily.

Response: Indeed, we manually checked up to 100 references per Boolean instruction (which would include Bliss synergy). If the result was higher than 100 hits, it is possible that we missed some publications: in those cases we looked for the word "combination index" which is present in 65% of the publications that describe synergy. This turned out to be the most effective way to collect a large amounts of publications with a relatively good recall. All Pubmed IDs and references as well as which method was used to determine the synergy have

now been added to supplementary table S1B-Curated-Data to enable an easy access to the data and comparison to the readers knowledge of published literature.

Detailed point 5: A much more convincing analysis would have looked at the recent DREAM challenge results, and worked directly with observed synergy data rather than the literature synergy determination above.

Response: We have analyzed the results of the DREAM challenge (Menden et al, 2019) in a similar fashion as for the Sanger data together with our literature curation. Statistical analysis showed significant and similar outcomes both for the distance concept as well as the sensitivity aspect as well as the performance in the prediction model. Given that the DREAM challenge contain substantial inter-assay noise (as noted by participants of the DREAM challenge), this might have weakened the correlations and/or predictive performance as compared to the Sanger data.

Detailed point 6: For the validation experiments, the choice of Chou&Talalay method for synergy determination is unnecessarily complex, and using the much simpler Bliss or HSA models would have found much the same results.

Response: Given that the combination index is most commonly used in the literature (see point 4), we used this method. Although the Chou&Talalay, Loewe, Bliss or HSA calculation methods have fundamental differences in the calculation of the synergy, they show overlap in the outcome of the calculation. We compared our results using all calculation methods and found a significant correlation between the combination index and the level of Loewe, BLISS and HSA synergy. Therefore, the following sentence was added to the supplemental methods: "The synergy as calculated by the Chou and Talalay Combination index method showed a significant correlation ($p < 0.0001$) to synergies calculated by either the Loewe, BLISS or HSA method."

Detailed point 7: The authors conflate the synergy observed in a cell line combination experiment with the increased durability of effect seen in their animal experiments. Those are two very different things - one relates to mechanistic interactions between targets in a cell, and the other relates to resistance that develops in a heterogeneous xenograft tumor. In fact, many combinations used in the clinic because they extend durable responses are not synergistic at all in the sense of a combination index. Indeed, a recent experiment (Palmer & Sorger 2017 Cell) showed that anti-resistance combinations rarely show any synergy whatsoever.

Response: (see also general statement 2) We appreciate this valuable comment and have now focused more on therapeutic synergy (which is highly relevant as can be concluded from the Palmer and Sorger 2017 paper) and have now changed the abstract and the interpretation of the in vivo results and have mentioned this relevant information in the introduction (around line 80):

"Many positive effects of drug combinations in the clinic are considered to be effective through the best response of a patient to either one of the two drugs (Palmer & Sorger 2017). Therefore, combinations of drugs are commonly more effective because each drug compensates for the drawback of the other drug. Currently, only a fraction of these combinations provide synergistic effects (Palmer & Sorger 2017). Therefore, the identification of such crucial mechanisms that can be targeted and lead to more than additive (synergistic) drug effects is therefore highly desirable."

Detailed point 8: The Discussion makes statements that are not well informed. The claim that drug synergies were uniformly considered to arise from "intimate process connections", overlooks that many in the literature have long assumed that distant targets would synergize better. In fact, synergies are known to happen in both situations, with different kinds of synergy: synthetic-lethal-like for parallel and potentiation-like for serial targets (Lehar et al. 2007 Mol Sys Bio). The synergy between Dasatinib and Imatinib is not at all surprising when you consider that they both have multiple targets (in the case of Dasatinib, many with comparable potency), which basically ensures that there are many targets involved in the interaction.

Response: We appreciate this valuable information and have now incorporated this in the discussion section (around line 350):

"Both closely related as well as unrelated processes have been considered accountable for drug synergies: they result from intimate process-connections (causing maximal target, pathway or feedback inhibition; Dancey,

2006; Léhar et al, 2007; Flaherty et al 2010; Dickson et al 2015; Ricordel et al, 2018) to less related parallel pathway connections (causing synthetic lethal interactions; Léhar et al, 2007; Srivas et al, 2016; Thomson et al, 2015, Shen and Ideker, 2018). Moreover, Gayvert et al, 2017 showed that synergistic combinations in mutant BRAF cell lines had a trend towards lower correlation of sensitivity over multiple cell lines, hence a drug-distance effect. Our method, that is based on common exclusive effects of drugs, is relevant for processes that are commonly only weakly connected. When these processes are simultaneously active in tumor cells, they offer a particular strong vulnerability given their independence. Our model therefore extend these previous findings, providing a new perspective on the relations of survival mechanisms.”.

Given the semantic problem with the term parallel (which could apply to within pathways as described by Léhar et al, 2007 as well as between pathways), the title of the manuscript is now changed to emphasize that we deal here with independent processes (i.e. between pathways). The Dasatinib and Imatinib example is now removed from the discussion although their off-targets are still mentioned in under the results section.

Detailed point 9: The manuscript reads like a rough draft. There are repeated statements, miss-spellings, and inaccurate figure references (where what’s described doesn’t match the figure). The color scales on some figures aren’t even consistent within one panel (see, eg. Fig.2G-viability).

Response: Repeated statements have been removed (mainly concerning the legends) except where they were functional. Part of Supplemental Figure S1 was accidentally indicated by S2. The color scale of Figure 2G has been made consistent.

Detailed point 10: There are many missing citations which are relevant to this work, both in terms of past combination screens that have been run, and methodologies used.

Response:

Added references according to previous points:

- Léhar et al, 2007
- Palmer & Sorger 2017

Reviewer #2 (Remarks to the Author):

General remark 1: Narayan et al showed a novel perspective on how to study drug sensitivity to inform the selection of drug combinations with synergistic effects in cancer treatment. The authors used publicly available data from various sources (2012 - 2018) to develop the computational approach and validate its predictions. The major claims of the paper are i) the drug distance obtained using the proposed drug atlas, together with drug sensitivity can be predictive of drug synergy; ii) the dual synergy can be utilized to predict the effectiveness of the synergy in triple drug combinations; iii) triple and dual drug combinations that showed synergy predicted by the proposed approach were validated in vivo.

There are various approaches to predict the synergy in pairwise drug combination screens however the multi-drug (N>3) combination synergy estimation is still lacking. This fact makes the paper of interest for cancer drug discovery. Another strength of the study is the experimental validation of the top ranked drug combinations in vivo. Overall, the proposed strategy is not complicated, which can allow for a simple extension in the lab or the clinic. The manuscript sections are well written and easy to follow, pictures are of good quality and readable. While the approach has merit, I think the authors have to improve some elements of the manuscript to increase its potential impact.

Response: We appreciate these positive comments of the reviewer.

General remark 2:

First of all, the manuscript has a poor description of the methods and techniques that authors applied for their study. The information from Supplemental Methods is not sufficient for making their work reproducible, and thus weaken the opportunity for other scientists to utilize the proposed strategy.

Response:

We have extended the Supplemental Methods, see highlighted text. In particular, we have added an extensive description of the data curation and the distance prediction model.

General remark 3: Second, there is a lack of formularization when it comes to the drug/target distances on the drug atlas. The authors used the cophenetic distance to show that the distance of synergistic drug pairs significantly exceeds the average distance between the drugs. However, when it comes to concrete dual or triple drug combinations that are used for validation, the authors refer to 'high distance drugs' without specifying the exact distances or any threshold they used to filter out small distance drugs. This information has to be clearly stated in the paper to avoid subjective drug/target distance evaluation.

Response: The prediction of synergy is based on the combined drug distance, sensitivity for both drugs as well as whether either one of drugs affect an oncogenic driver. Therefore it turned out to be difficult to provide a strict formularization of the distance that provides the strongest synergy since its optimal value is determined by the other parameters. In addition, the distance space between drugs is not linear (see Figure 2A-C, S2A, B, right part of each figure for distribution) which is to be taken into account as well. The drug combinations for the triple-drug experiments were based on a number of drug pairs that commonly showed large distances as well as a relative high sensitivity (oncogenic targeting did not play a prominent role except for SF-268 containing the A289V EGFR mutation) but were not strictly chosen for a large distance. The point of the triple therapy section is to show that double synergies can guide the selection triple synergies which is indeed valuable for drug discovery/polypharmacology.

General remark 4: Finally, the weakest point of the paper is the proposed synergy prediction model, as there is a poor description of the model, no information present about accuracy evaluation in the model training and no data available to check the rankings of the drug combinations based on the model.

Response: We wrote the description of the synergy prediction model together with a statistician (RM) and provide information on the performance, also on the DREAM benchmark data of Menden et al, 2019. Rankings are provided in Table S4A.

Major comment 1: Given that only small molecule drugs were used in the studies, the authors should state whether the proposed approach is limited to those or justify that it can be extended for other drugs (like immunotherapeutics and perhaps antibody drugs).

Response: Because each drug that was used to create the drug atlas had annotated targets, we were able to define target-distances and target-sensitivities. This step makes it possible to extrapolate the drug atlas to drugs that are not part of the drug atlas, provided that the targets of these drugs are known. So, for instance, for an antibody with a known target (matching a drug atlas target), it is possible to make a prediction whether this antibody will show synergy with another drug.

Major comment 2: The authors collected drug response data from GDSC and CCLE. However, there were concerns about the reproducibility of the drug screen studies (see e.g. PMID 24284626). The authors are advised to confirm that these multiple data sources are in good consistency.

Response: The drug atlas is based on the drug action of the first generation of the GDSC data (MGH subset) that was selected because these drugs showed the most consistent clustering (see also S1F, blue colored drugs). The text of the manuscript was changed to clarify this (around line 155).

Major comment 3: In the Supplemental Methods the authors describe the way how they generated the drug atlas. However, it can be done in a better way, so that a reader that is not familiar with the listed methods can understand the main logic behind them. Thus, the description for such techniques as calculating branch grouping threshold, weighted FVL and map smoothing has to be improved.

Response: Detail regarding the drug atlas Voronoi methodology are now added to the Supplemental Materials and Methods.

Major comment 4: It is not clear how synergistic interactions were mapped to the drug atlas and thus visualized in figures S1E and S1F.

Response: This list of drug synergies as obtained by the literature curation Published synergistic drug pairs are shown in Figure S1E as “Published synergistic drug pairs” and “Published synergy between targets”. Perhaps confusingly, S1F does not provide any synergy information but provides a comparison between different generations of the GDSC datasets. Therefore we added “Comparison datasets” above the Voronoi diagram.

Major comment 5: In order to obtain ground-truth data for the approach, the authors performed literature survey. They included papers that fulfilled their searching criteria. However, there is no information provided what thresholds were applied on different synergy metrics, whether those were consistent within the papers, were there cases of overlapping synergistic drug combinations in papers (how the authors evaluate synergy in this case), were there cases where drug combinations showed controversial scores within different papers.

Response: For synergy, there is no formal threshold other than that the combined effect should be more than the sum of each effect. The criteria that we used were:

- (1) Combination Index Chou and Talalay <1.0
 - (2) the t-test/histogram analysis should be significant $p < 0.05$
 - (3) the isobologram should show a combined effect below the additive effect
- For other variants of assessing the synergy, the similar thresholds were used.

Major comment 6: In the drug-drug synergy experiments the authors mention that per drug the start concentration was determined based on the IC50 curves. It will be useful to know how exactly the concentrations were chosen for the experiments: whether they had to span uniformly around the IC50 values or the starting concentrations were close to IC50 only.

It is also stated in the paper that only 3 concentrations were utilized in the drug-drug experiments. Is this sufficient to capture the whole scope of the treatment response? This question again refers to the rationale of the choice of the concentrations that was not clearly stated in the paper.

Response: These points are now changed and clarified in the supplemental methods, the section was accidentally copied from the triple synergy methodology. For clarity, for each drug combination, a matrix of 36 combinations was used (five concentrations plus a blank for each drug) and each experiment was measured in triplicate. For the triple combinations we used initially a similar setup but because of the enormous scale of each experiments (36×6 [layers] \times 3 [triplicate]=648 wells per assay), confirmed a number of synergies in a smaller 3×3 matrix.

Major comment 7. The authors used the median effect principle by Chou and Tallalay to calculate CI. Have they tried other metrics for comparison (e.g. Bliss mode and ZIP model, PMID 26949479). It will be also interesting to see the quantitative difference between the primary CI and secondary CI.

Response: (1) Given that the combination index is most commonly used in the literature (65%), we used this method. We compared the commonly used Chou&Talalay, Loewe, Bliss or HSA calculation methods and found a significant correlation between the combination index and the synergy level of these methods. Therefore, the following sentence was added to the supplemental methods: “The synergy as calculated by the Chou and Talalay Combination index method showed a significant correlation ($p < 0.0001$) to synergies calculated by either the Loewe, BLISS or HSA method.” Given this exhaustive comparison with the methods that are most commonly used in the literature, we hope that the reviewer is convinced that this subject is sufficiently covered and is likely to match the ZIP model as well given the agreement between independent methods. We added the primary CI to the supplemental data (Table S3).

Major comment 8. It is unclear how the multiple linear regression model training was performed (how did they estimate the accuracy of the model?) as well as how exactly the weights s , d , t , c were chosen. The log-rank test for prioritizing drug combinations has to be also explained. The authors can provide an xlsx file containing the rankings for the combinations.

Response: We have updated the description of the model with the help of a statistician. The weights of the covariates s , d , t and c were estimated using the logistic model.

Major comment 9. The authors refer to 'high distance drugs' when performing experimental validation. A formal

definition should be created to be able to separate high and small distance drugs. It might be helpful to see the table or a plot where sensitivity/synergy of drug combinations is compared to the corresponding drug distances from the drug atlas.

Response: The prediction of synergy is based on the combined drug distance, sensitivity for both drugs as well as whether either one of drugs affect an oncogenic driver. Therefore it turned out to be difficult to provide a strict formularization of the distance that provides the strongest synergy since its optimal value is determined by the other parameters. In addition, the distance space between drugs is not linear (see Figure 2A-C, S2A, B, right part of each figure for distribution) which is to be taken into account as well.

Detailed minor points 1: In the Figure 3C description, there is (D) character instead of (C). The description does not correspond to the actual Figure either (it is a correlation plot, what is r value ?)

Response: Error is corrected, r stands for Spearman correlation which is now added to the legend.

Detailed minor points 2: Line 256 use 'too' instead of 'to'.

Response: Spelling error is corrected.

Detailed minor points 3: Line 243 in Supplemental Methods - remove one extra 'distance'.

Response: Error was corrected.

Reviewers' comments:

Reviewer #2 (Remarks to the Author):

I appreciate that Narayan et al revised the manuscript significantly. However, there are some major comments that have to be still clarified by the authors.

Major comment 2: My original comment was asking about the consistency of the same drug-cell pair across different studies. For example, if a same drug-cell line was tested both in Sanger and in CCLE how these 'replicates' were treated? The authors decided to use only the Sanger (MGH) data. However, the rational of using clustering to determine the data quality is not clear.

Furthermore, the clustering on CCLE data was not shown in Figure S1F.

Major comment 3: what is the software tool that was used to generate the weighted Voronoi diagram? If no existing software is available then the source code for generating the graph need to be attached to allow readers to test and apply this method easily.

Major comment 5: the authors mentioned the criteria that they used for annotating the drug combination as synergistic, however, they had not added this important information to the manuscript. The authors stated that they identified a drug combination as synergistic if the combined effect was greater than the sum of single drug effects. However, that fact means that another, less strict metrics (e.g. Bliss, as it has additive term to the difference between the combined and sum of single effects), can recognize the drug combination as synergistic even if the author's criterion is not fulfilled. Besides, the last part of the comment was not addressed.

Major comment 7: The authors compared the CI method with other synergy models and confirmed a high correlation. However, some figures or a supplementary table with all the metrics will support the results much better than one additional sentence given that the results have been already obtained.

Major comment 8: The authors added a detailed description of the prediction model and summarized the results of the drug distance and sensitivity contribution using the AUC of the dose response curve. It will be interesting to see whether the findings are similar for all the cancer types or drug classes or if there are some exceptions for which adding drug distance and/or sensitivity information do not lead to a better synergy prediction.

New Major comment A: How the drug targets were determined before calculating the target distance? Many compounds are known to be polypharmacological and determining the target profile is not trivial.

New Major comment B: how the drug atlas distance was determined on the DREAM data? It was described in Supplementary File that drug distance was calculated as 1 minus the similarity between the AUC data vectors. However I understood that DREAM data did not provide AUC-based single drug sensitivity. Furthermore, how the training and testing data was split in the logistic regression model was unclear.

Minor comments:

1. The definition of cophenetic distance is unclear. A mathematical formula would help.
2. ROC curve should be Receiver operating characteristic curve, not 'responder operator curve'.
3. It is advised that the authors also provide a clean version of the manuscript for a better readability.

Reviewer #3 (Remarks to the Author):

Narayanan et al present an interesting approach to predict synergistic combination therapies from drug-vulnerability based on mono-therapy data. The authors have done an overall good job to address the reviewer comments, which have substantially improved the manuscript.

However, the following points were not sufficiently addressed and need to be addressed more carefully, before the manuscript can be considered for publication.

- Reviewer #1 (Detailed point 3 and 6) and Reviewer #2 (Major comment 7) raised the point, that whether synergy is found or not might depend on the underlying additivity criterion. In Detailed Point 6, Reviewer #1 criticized that the authors relied on the Chou and Talalay method for synergy determination in the validation dataset, which was claimed by Reviewer #1 to be unnecessarily complex. While I agree that Bliss and HSA are simpler, I think the more important aspect is how the different methods relate. This was only partly addressed by the authors. While a correlation between Bliss and CI (i.e. usually Loewe Additivity) is found, the nature of the determined interaction (i.e. if synergy or antagonism is concluded) can be different under Bliss or Loewe. Please report also if the interaction types were consistent across the evaluated methods, not only the correlation.

- Reviewer #2, major comment 1: I feel, this aspect would deserve to be also addressed in the Discussion section of the manuscript.

- A limitation of the study is that synergy was not evaluated in a mechanistic way. In fact, models were recently developed that can distinguish between interactions on potency as well as maximum effect and even inform on directionality in a drug interaction, i.e. when interactions are concentration-dependent (e.g. PMID 29242552 published earlier in this journal). It would be interesting to study, whether the drug distance relates to a distinct pattern on the effect level, when the joint response surface would be analyzed by a more sophisticated model. The authors may want to address this limitation of their work and/or add this aspect as a perspective.

Reviewer # 2

(Remarks to the Author) “I appreciate that Narayan et al revised the manuscript significantly. However, there are some major comments that have to be still clarified by the authors”.

Major comment 2: My original comment was asking about the consistency of the same drug-cell pair across different studies. For example, if a same drug-cell line was tested both in Sanger and in CCLE how these ‘replicates’ were treated? The authors decided to use only the Sanger (MGH) data. However, the rationale of using clustering to determine the data quality is not clear. Furthermore, the clustering on CCLE data was not shown in Figure S1F.

Response: The overlap in drugs between different datasets with experimental drugs is commonly very small, given the enormous amount of available preclinical compounds. This was particularly true for the GDSC data, containing MGH and WTSI datasets described together by Garnett et al, pubmed 22460902, where the overlap was only one drug (i.e. Camptothecin). There is an overlap of 9 drugs between GDSC and the CCLE datasets (Safikhani et al, pubmed 28928933), however, since the CCLE dataset contains only 24 drugs with only 9 drugs overlap with the GDSC dataset, we could not generate a proper drug-atlas from this and it would be hard to imagine that this would improve the GDSC drug atlas. Nevertheless, we have tried to integrate data of different sources. Clustering of dose response data of the GDSC data (i.e. the WTSI and MGH data and a later update) indicates that the research institute or versions of the data (i.e. experimental conditions) rather than the response data explain the clustering. This indicates that there are systematic differences between the institutes which warrant against combining datasets.

Major comment 3: what is the software tool that was used to generate the weighted Voronoi diagram? If no existing software is available then the source code for generating the graph need to be attached to allow readers to test and apply this method easily.

Response: Given that the Voronoi methodology is still developing and is not yet at the stage where it can be published as a proper Cytoscape app, we can provide the software to the reviewer through a restricted access repository in Github. In the future the software will be shared upon a reasonable request (this is now added to the supplemental materials and methods).

Major comment 5: the authors mentioned the criteria that they used for annotating the drug combination as synergistic, however, they had not added this important information to the manuscript. The authors stated that they identified a drug combination as synergistic if the combined effect was greater than the sum of single drug effects. However, that fact means that another, less strict metrics (e.g. Bliss, as it has additive term to the difference between the combined and sum of single effects), can recognize the drug combination as synergistic even if the author’s criterion is not fulfilled. Besides, the last part of the comment was not addressed (i.e. “were there cases where drug combinations showed controversial scores within different papers.”).

Response: We agree with the reviewer that the different models to describe synergy, particularly for Bliss additivity, make it difficult to compare experimental outcomes (Vlot et al PMID: 31518641). However, here this problems does not seem to be so important since 88% of the papers mention the combination index as applied metric. Bliss additivity was mentioned in only two papers, i.e. 1% of the literature benchmark data. To make this point we added the sentence around line 146: “; Synergy metrics: combination index was present in 88% of the cases, less strict Bliss additivity was present in 1% of the identified cases)”.

Major comment 7: The authors compared the CI method with other synergy models and confirmed a high correlation. However, some figures or a supplementary table with all the metrics will support the results much better than one additional sentence given that the results have been already obtained.

Response: We now provide the comparison of the CI to other synergy models in a supplemental figure (S2G and H), see also reviewer 3 point 1. We also determined whether the synergy label was concordant (antagonism was occasionally observed and therefore neglected), see Figure S2H. This shows that strong synergy and absence of synergy are commonly concordant but that weak interactions provide different Loewe or Bliss additivity outcomes. This is now also mentioned in the text of the manuscript: “The synergy as calculated by the Chou and Talalay Combination index method showed a significant correlation ($p < 0.0001$) to synergies calculated by either the Loewe, BLISS or HSA method (Figure S2G and Supplemental Table S2C, where each model interprets weak interactions differently, Figure S2H).”

Major comment 8: The authors added a detailed description of the prediction model and summarized the results of the drug distance and sensitivity contribution using the AUC of the dose response curve. It will be interesting to see whether the findings are similar for all the cancer types or drug classes or if there are some exceptions for which adding drug distance and/or sensitivity information do not lead to a better synergy prediction.

Response: We have now added this information to the manuscript: around line 227: “Synergies were relatively more often observed in the literature and predicted for breast cancer and less often observed and predicted for lung tumors (both $p < 0.00001$; Figure S2H).”

New Major comment A: How the drug targets were determined before calculating the target distance? Many compounds are known to be polypharmacological and determining the target profile is not trivial.

Response: The drug atlas was originally designed to incorporate a polypharmacology mechanism of action of drugs since the drug distance is expected to reflect the sum of on-target as well as off-target mechanisms. However, to connect the drug distance and sensitivity to other (non-overlapping) drugs for the benchmarking, we had to rely on the primary target of the drug (GDSC defined targets). We hope that it is clear that this issue is only present when benchmarking the atlas with external data, where there is no overlap in the applied combinations of drugs (which is often the case given that missing data expand exponentially in drug combinations).

New Major comment B: how the drug atlas distance was determined on the DREAM data? It was described in Supplementary File that drug distance was calculated as 1 minus the similarity between the AUC data vectors. However I understood that DREAM data did not provide AUC-based single drug sensitivity. Furthermore, how the training and testing data was split in the logistic regression model was unclear.

Response: In order to validate the predictive value of the drug atlas, we used the drug atlas based prediction model to predict synergies of the DREAM data. Therefore, it was unnecessary to build a new model. As a remark, the single-drug sensitivities of the DREAM data are not particularly well suited for an AUC determination and since the data also showed inter-experimental inconsistencies, it would have been difficult to generate a prediction model based on this data.

Minor comments:

1. The definition of cophenetic distance is unclear. A mathematical formula would help.
2. ROC curve should be Receiver operating characteristic curve, not 'responder operator curve'.
3. It is advised that the authors also provide a clean version of the manuscript for a better readability.

Response:

1. Formula is added to the supplemental materials and methods.
2. Spelling of ROC was corrected.
3. We will provide a version of the manuscript with accepted changes.

Reviewer #3

(Remarks to the Author) "Narayan et al present an interesting approach to predict synergistic combination therapies from drug-vulnerability based on mono-therapy data. The authors have done an overall good job to address the reviewer comments, which have substantially improved the manuscript.

However, the following points were not sufficiently addressed and need to be addressed more carefully, before the manuscript can be considered for publication."

Reviewer #1 (Detailed point 3 and 6) and Reviewer #2 (Major comment 7) raised the point, that whether synergy is found or not might depend on the underlying additivity criterion. In Detailed Point 6, Reviewer #1 criticized that the authors relied on the Chou and Talalay method for synergy determination in the validation dataset, which was claimed by Reviewer #1 to be unnecessarily complex. While I agree that Bliss and HSA are simpler, I think the more important aspect is how the different methods relate. This was only partly addressed by the authors. While a correlation between Bliss and CI (i.e. usually Loewe Additivity) is found, the nature of the determined interaction (i.e. if synergy or antagonism is concluded) can be different under Bliss or Loewe. Please report also if the interaction types were consistent across the evaluated methods, not only the correlation.

Response: We now provide the comparison of the CI to other synergy models in a supplemental figure (S2G and H), see reviewer 2 point 7. We also determined whether the synergy label was concordant (antagonism was occasionally observed and therefore neglected), see Figure S2H. This shows that strong synergy and absence of synergy are commonly concordant but that weak interactions provide different Loewe or Bliss additivity outcomes. This is now also mentioned in the text of the manuscript: "The synergy as calculated by the Chou and Talalay Combination index method showed a significant correlation ($p < 0.0001$) to synergies calculated by either the Loewe, BLISS or HSA method (Figure S2G and Supplemental Table S2C, where each model interprets weak interactions differently, Figure S2H)."

Reviewer #2, major comment 1: I feel, this aspect would deserve to be also addressed in the Discussion section of the manuscript. Citation major remark 1: "Given that only small molecule drugs were used in the studies, the authors should state whether the proposed approach is limited to those or justify that it can be extended for other drugs (like immunotherapeutics and perhaps antibody drugs)"

Response: We have now added the following sentence to the discussion: "...multi-drug combinations which could increase drug combination efficacy, reduce therapy resistance and could aid in designing optimal polypharmacological (i.e. multi-targeted) therapies, both for small molecules as well as antibody driven approaches." We also added the sentence: "Unfortunately, we were not able as yet to connect the drug atlas to various immunotherapies, which is a goal for the near future."

Reviewer #3, point 1: A limitation of the study is that synergy was not evaluated in a mechanistic way. In fact, models were recently developed that can distinguish between interactions on potency as well as maximum effect and even inform on directionality in a drug interaction, i.e. when interactions are concentration-dependent (e.g. PMID 29242552 published earlier in this journal). It would be interesting to study, whether the drug distance relates to a distinct pattern on the effect level, when the joint response surface would be analyzed by a more sophisticated model. The authors may want to address this limitation of their work and/or add this aspect as a perspective.

Response: Around line 210 we added the sentence "improvement of the interpretation of these interaction data could become more robust by taking more complex interactions into account according to Wicha et al, 2017". Around line 408 in the discussion we added "In the future, the drug atlas prediction model might be coupled to a more sophisticated general pharmacodynamic interaction (GPDI) model of synergy (Wicha et al, 2017) and thereby refine the applied strategies as provided here."

Appendix 1. Previous, not followed up, review report of Reviewer 1

Reviewer #1 (Remarks to the Author):

General remark 1: The core concept behind the proposed method is promising. Basically that the best synergies will occur between drugs targeting very different mechanisms, as revealed by how they affect many cell lines. This idea is not completely novel, especially because most cancer combination regimens specifically focus on diverse mechanisms, in order to overcome drug resistance, but the drug response atlas does neatly capture a way to identify useful combinations.

Response: We appreciate these positive comments of the reviewer.

General statement 2: However, the execution of this idea and the validation experiments are flawed. Moreover, the authors conflate the synergy observed in an cell line combination experiment with the increased durability of effect seen in their animal experiments. The result is that the logical connections between drug-response -> synergy -> in-vivo response are just not convincing.

Response: Due to this critical note we now removed the implicated link between in vitro synergy and durability of the response. The experimental results do not (sufficiently) support this and distract from the relevant observation that synergy is actually measured in vivo. A clearer explanation and provisions are now added to the manuscript. Accordingly we changed this throughout the manuscript.

Detailed point 1: The drug atlas generation is a nice way of comparing drug response profiles across all the cell lines. This is a novel approach that very nicely organizes drug pairs in a way that will be informative.

Response: We appreciate this positive remark.

Detailed point 2: The literature synergy data generation is very unconvincing. Searching for CellName+"synerg*"+"Combination Index" will provide very misleading results. Many cell lines are known by differing descriptors, and the authors haven't described how they addressed that problem. Synergy as a term could relate to what was found or what was not found.

Response: Thank you for this critical note. The given description in the materials and methods was too brief. We have now provided the full methodology that was used to identify the references which is now correctly described in the supplemental methods and supplemental Table S1B (Curated data), which shows which method that was used to determine the synergy for each case.

Detailed point 3: Also searching for Combination Index will only find synergies using the Chou+Talalay approach, so it's unclear how the authors came up with combinations listed as Bliss synergies.

Response: This points is now corrected, see the previous point.

Detailed point 4: Moreover, how the authors set their cutoff of "~100 references" for a combination, and curated their hits is not described. I'm guessing that much of the curation was manual inspection of many manuscripts, which probably explains why so few "synergy" calls (244) are determined. Some combinations known to synergize in these cell lines do not appear to be in the list. It would be very helpful if the supplementary table S1B-Curated-Data put the reference alongside each hit, so that it could be checked more easily.

Response: Indeed, we manually checked up to 100 references per Boolean instruction (which would include Bliss synergy). If the result was higher than 100 hits, it is possible that we missed some

publications: in those cases we looked for the word “combination index” which is present in 65% of the publications that describe synergy. This turned out to be the most effective way to collect a large amounts of publications with a relatively good recall. All Pubmed IDs and references as well as which method was used to determine the synergy have now been added to supplementary table S1B-Curated-Data to enable an easy access to the data and comparison to the readers knowledge of published literature.

Detailed point 5: A much more convincing analysis would have looked at the recent DREAM challenge results, and worked directly with observed synergy data rather than the literature synergy determination above.

Response: We have analyzed the results of the DREAM challenge (Menden et al, 2019) in a similar fashion as for the Sanger data together with our literature curation. Statistical analysis showed significant and similar outcomes both for the distance concept as well as the sensitivity aspect as well as the performance in the prediction model. Given that the DREAM challenge contain substantial inter-assay noise (as noted by participants of the DREAM challenge), this might have weakened the correlations and/or predictive performance as compared to the Sanger data.

Detailed point 6: For the validation experiments, the choice of Chou&Talalay method for synergy determination is unnecessarily complex, and using the much simpler Bliss or HSA models would have found much the same results.

Response: Given that the combination index is most commonly used in the literature (see point 4), we used this method. Although the Chou&Talalay, Loewe, Bliss or HSA calculation methods have fundamental differences in the calculation of the synergy, they show overlap in the outcome of the calculation. We compared our results using all calculation methods and found a significant correlation between the combination index and the level of Loewe, BLISS and HSA synergy. Therefore, the following sentence was added to the supplemental methods: “The synergy as calculated by the Chou and Talalay Combination index method showed a significant correlation ($p < 0.0001$) to synergies calculated by either the Loewe, BLISS or HSA method.”

Detailed point 7: The authors conflate the synergy observed in a cell line combination experiment with the increased durability of effect seen in their animal experiments. Those are two very different things - one relates to mechanistic interactions between targets in a cell, and the other relates to resistance that develops in a heterogeneous xenograft tumor. In fact, many combinations used in the clinic because they extend durable responses are not synergistic at all in the sense of a combination index. Indeed, a recent experiment (Palmer & Sorger 2017 Cell) showed that anti-resistance combinations rarely show any synergy whatsoever.

Response: (see also general statement 2) We appreciate this valuable comment and have now focused more on therapeutic synergy (which is highly relevant as can be concluded from the Palmer and Sorger 2017 paper) and have now changed the abstract and the interpretation of the in vivo results and have mentioned this relevant information in the introduction (around line 80):

“Many positive effects of drug combinations in the clinic are considered to be effective through the best response of a patient to either one of the two drugs (Palmer & Sorger 2017). Therefore, combinations of drugs are commonly more effective because each drug compensates for the drawback of the other drug. Currently, only a fraction of these combinations provide synergistic effects (Palmer

& Sorger 2017). Therefore, the Identification identification of such crucial mechanisms that can be targeted and lead to more than additive (synergistic) drug effects is therefore highly desirable.”

Detailed point 8: The Discussion makes statements that are not well informed. The claim that drug synergies were uniformly considered to arise from “intimate process connections”, overlooks that many in the literature have long assumed that distant targets would synergize better. In fact, synergies are known to happen in both situations, with different kinds of synergy: synthetic-lethal-like for parallel and potentiation-like for serial targets (Lehar et al. 2007 Mol Sys Bio). The synergy between Dasatinib and Imatinib is not at all surprising when you consider that they both have multiple targets (in the case of Dasatinib, many with comparable potency), which basically ensures that there are many targets involved in the interaction.

Response: We appreciate this valuable information and have now incorporated this in the discussion section (around line 350):

“Both closely related as well as unrelated processes have been considered accountable for drug synergies: they result from intimate process-connections (causing maximal target, pathway or feedback inhibition; Dancey, 2006; Léhar et al, 2007; Flaherty et al 2010; Dickson et al 2015; Ricordel et al, 2018) to less related parallel pathway connections (causing synthetic lethal interactions; Léhar et al, 2007; Srivas et al, 2016; Thomson et al, 2015, Shen and Ideker, 2018). Moreover, Gayvert et al, 2017 showed that synergistic combinations in mutant BRAF cell lines had a trend towards lower correlation of sensitivity over multiple cell lines, hence a drug-distance effect. Our method, that is based on common exclusive effects of drugs, is relevant for processes that are commonly only weakly connected. When these processes are simultaneously active in tumor cells, they offer a particular strong vulnerability given their independence. Our model therefore extend these previous findings, providing a new perspective on the relations of survival mechanisms.”.

Given the semantic problem with the term parallel (which could apply to within pathways as described by Léhar et al, 2007 as well as between pathways), the title of the manuscript is now changed to emphasize that we deal here with independent processes (i.e. between pathways). The Dasatinib and Imatinib example is now removed from the discussion although their off-targets are still mentioned in under the results section.

Detailed point 9: The manuscript reads like a rough draft. There are repeated statements, misspellings, and inaccurate figure references (where what’s described doesn’t match the figure). The color scales on some figures aren’t even consistent within one panel (see, eg. Fig.2G-viability).

Response: Repeated statements have been removed (mainly concerning the legends) except where they were functional. Part of Supplemental Figure S1 was accidentally indicated by S2. The color scale of Figure 2G has been made consistent.

Detailed point 10: There are many missing citations which are relevant to this work, both in terms of past combination screens that have been run, and methodologies used.

Response:

Added references according to previous points:

- Léhar et al, 2007
- Palmer & Sorger 2017

REVIEWERS' COMMENTS:

Reviewer #2 (Remarks to the Author):

The authors did not convincingly solve my concerns in the previous rounds of comments. Inclusion of the DREAM challenge data made the conclusion weaker and therefore the general applicability of the models is questionable.

Reviewer #3 (Remarks to the Author):

All comments were adequately addressed. Thank you!

Response to Reviewers

REVIEWERS' COMMENTS:

Reviewer #2 (Remarks to the Author):

Remark: The authors did not convincingly solve my concerns in the previous rounds of comments. Inclusion of the DREAM challenge data made the conclusion weaker and therefore the general applicability of the models is questionable.

Response: The DREAM data show a strong overlap with the GDSC based data and proof that the model can be applied to external data. All other previous responses are listed in **Appendix 1** and, to our best belief, address all raised points of the reviewer.

Reviewer #3 (Remarks to the Author):

Remark: All comments were adequately addressed. Thank you!

Response: no more comments

Appendix 1 Previous responses to reviewer 2

Reviewer # 2

(Remarks to the Author) “I appreciate that Narayan et al revised the manuscript significantly. However, there are some major comments that have to be still clarified by the authors”.

Major comment 2: My original comment was asking about the consistency of the same drug-cell pair across different studies. For example, if a same drug-cell line was tested both in Sanger and in CCLE how these ‘replicates’ were treated? The authors decided to use only the Sanger (MGH) data. However, the rationale of using clustering to determine the data quality is not clear. Furthermore, the clustering on CCLE data was not shown in Figure S1F.

Response: The overlap in drugs between different datasets with experimental drugs is commonly very small, given the enormous amount of available preclinical compounds. This was particularly true for the GDSC data, containing MGH and WTSI datasets described together by Garnett et al, pubmed 22460902, where the overlap was only one drug (i.e. Camptothecin). There is an overlap of 9 drugs between GDSC and the CCLE datasets (Safikhani et al, pubmed 28928933), however, since the CCLE dataset contains only 24 drugs with only 9 drugs overlap with the GDSC dataset, we could not generate a proper drug-atlas from this and it would be hard to imagine that this would improve the GDSC drug atlas. Nevertheless, we have tried to integrate data of different sources. Clustering of dose response data of the GDSC data (i.e. the WTSI and MGH data and a later update) indicates that the research institute or versions of the data (i.e. experimental conditions) rather than the response data explain the clustering. This indicates that there are systematic differences between the institutes which warrant against combining datasets.

Major comment 3: what is the software tool that was used to generate the weighted Voronoi diagram? If no existing software is available then the source code for generating the graph need to be attached to allow readers to test and apply this method easily.

Response: Given that the Voronoi methodology is still developing and is not yet at the stage where it can be published as a proper Cytoscape app, we can provide the software to the reviewer through a restricted access repository in Github. In the future the software will be shared upon a reasonable request (this is now added to the supplemental materials and methods).

Major comment 5: the authors mentioned the criteria that they used for annotating the drug combination as synergistic, however, they had not added this important information to the manuscript. The authors stated that they identified a drug combination as synergistic if the combined effect was greater than the sum of single drug effects. However, that fact means that another, less strict metrics (e.g. Bliss, as it has additive term to the difference between the combined and sum of single effects), can recognize the drug combination as synergistic even if the author’s criterion is not fulfilled. Besides, the last part of the comment was not addressed (i.e. “were there cases where drug combinations showed controversial scores within different papers.”).

Response: We agree with the reviewer that the different models to describe synergy, particularly for Bliss additivity, make it difficult to compare experimental outcomes (Vlot et al PMID: 31518641). However, here this problems does not seem to be so important since 88% of the papers mention the combination index as applied metric. Bliss additivity was mentioned in only two papers, i.e. 1% of the literature benchmark data. To make this point we added the sentence around line 146: “;

Synergy metrics: combination index was present in 88% of the cases, less strict Bliss additivity was present in 1% of the identified cases)”.

Major comment 7: The authors compared the CI method with other synergy models and confirmed a high correlation. However, some figures or a supplementary table with all the metrics will support the results much better than one additional sentence given that the results have been already obtained.

Response: We now provide the comparison of the CI to other synergy models in a supplemental figure (S2G and H), see also reviewer 3 point 1. We also determined whether the synergy label was concordant (antagonism was occasionally observed and therefore neglected), see Figure S2H. This shows that strong synergy and absence of synergy are commonly concordant but that weak interactions provide different Loewe or Bliss additivity outcomes. This is now also mentioned in the text of the manuscript: “The synergy as calculated by the Chou and Talalay Combination index method showed a significant correlation ($p < 0.0001$) to synergies calculated by either the Loewe, BLISS or HSA method (Figure S2G and Supplemental Table S2C, where each model interprets weak interactions differently, Figure S2H).”

Major comment 8: The authors added a detailed description of the prediction model and summarized the results of the drug distance and sensitivity contribution using the AUC of the dose response curve. It will be interesting to see whether the findings are similar for all the cancer types or drug classes or if there are some exceptions for which adding drug distance and/or sensitivity information do not lead to a better synergy prediction.

Response: We have now added this information to the manuscript: around line 227: “Synergies were relatively more often observed in the literature and predicted for breast cancer and less often observed and predicted for lung tumors (both $p < 0.00001$; Figure S2H).”

New Major comment A: How the drug targets were determined before calculating the target distance? Many compounds are known to be polypharmacological and determining the target profile is not trivial.

Response: The drug atlas was originally designed to incorporate a polypharmacology mechanism of action of drugs since the drug distance is expected to reflect the sum of on-target as well as off-target mechanisms. However, to connect the drug distance and sensitivity to other (non-overlapping) drugs for the benchmarking, we had to rely on the primary target of the drug (GDSC defined targets). We hope that it is clear that this issue is only present when benchmarking the atlas with external data, where there is no overlap in the applied combinations of drugs (which is often the case given that missing data expand exponentially in drug combinations).

New Major comment B: how the drug atlas distance was determined on the DREAM data? It was described in Supplementary File that drug distance was calculated as 1 minus the similarity between the AUC data vectors. However I understood that DREAM data did not provide AUC-based single drug sensitivity. Furthermore, how the training and testing data was split in the logistic regression model was unclear.

Response: In order to validate the predictive value of the drug atlas, we used the drug atlas based prediction model to predict synergies of the DREAM data. Therefore, it was unnecessary to build a new model. As a remark, the single-drug sensitivities of the DREAM data are not particularly well suited for an AUC determination and since the data also showed inter-experimental inconsistencies, it would have been difficult to generate a prediction model based on this data.

Minor comments:

1. The definition of cophenetic distance is unclear. A mathematical formula would help.
2. ROC curve should be Receiver operating characteristic curve, not 'responder operator curve'.
3. It is advised that the authors also provide a clean version of the manuscript for a better readability.

Response:

1. Formula is added to the supplemental materials and methods.
2. Spelling of ROC was corrected.
3. We will provide a version of the manuscript with accepted changes.